# Rare variant contribution to human disease in 281,104 UK Biobank exomes

Quanli Wang[1,20], Ryan S. Dhindsa[1,20], Keren Carss[2,20], Andrew R. Harper[2], Abhishek Nag[2], Ioanna Tachmazidou[2], Dimitrios Vitsios[2], Sri V. V. Deevi[2], Alex Mackay[3], Daniel Muthas[3], Michael Hühn[3], Susan Monkley[3], Henric Olsson[3], AstraZeneca Genomics Initiative*, Sebastian Wasilewski[2], Katherine R. Smith[2], Ruth March[4], Adam Platt[5], Carolina Haefliger[2] & Slavé Petrovski[2,6,7 ✉]

Genome-wide association studies have uncovered thousands of common variants associated with human disease, but the contribution of rare variants to common disease remains relatively unexplored. The UK Biobank contains detailed phenotypic data linked to medical records for approximately 500,000 participants, offering an unprecedented opportunity to evaluate the effect of rare variation on a broad collection of traits[1,2]. Here we study the relationships between rare protein-coding variants and 17,361 binary and 1,419 quantitative phenotypes using exome sequencing data from 269,171 UK Biobank participants of European ancestry. Gene-based collapsing analyses revealed 1,703 statistically significant gene–phenotype associations for binary traits, with a median odds ratio of 12.4. Furthermore, 83% of these associations were undetectable via single-variant association tests, emphasizing the power of gene-based collapsing analysis in the setting of high allelic heterogeneity. Gene–phenotype associations were also significantly enriched for loss-of-function-mediated traits and approved drug targets. Finally, we performed ancestry-specific and pan-ancestry collapsing analyses using exome sequencing data from 11,933 UK Biobank participants of African, East Asian or South Asian ancestry. Our results highlight a significant contribution of rare variants to common disease. Summary statistics are publicly available through an interactive portal (http://azphewas.com/).

The identification of genetic variants that contribute to human disease has facilitated the development of highly efficacious and safe therapeutic agents[3–5]. Drug candidates targeting genes with evidence of human disease causality are in fact substantially more likely to be approved[6,7]. Exome sequencing has revolutionized our understanding of rare diseases, uncovering causal rare variants for hundreds of these disorders. However, most efforts for complex human diseases and traits have relied on genome-wide association studies (GWAS), which focus on common variants. Compared with rare variants, common variants tend to confer smaller effect sizes and can be difficult to map to causal genes[8].

The UK Biobank (UKB) offers an unprecedented opportunity to assess the contribution of both common and rare genetic variation to thousands of human traits and diseases[1,2,9–13]. Testing for the association between rare variants and phenotypes is typically performed at the variant or gene level. Gene-level association tests include collapsing analyses and burden tests, among others[14–17]. Collapsing analyses are particularly well suited to detect genetic risk for phenotypes driven by an allelic series[16–23] and can provide a clear link between the causal gene and phenotype. Applications of these methods to the first 50,000 UKB exome sequences have indicated an important role of rare variation in complex disease but have also highlighted a need for larger sample sizes[10,11].

In this study, we performed a phenome-wide association study (PheWAS) using exome sequence data from 269,171 UKB participants of European ancestry to evaluate the association between protein-coding variants and 17,361 binary and 1,419 quantitative phenotypes. We first report the diversity of phenotypes and sequence variation present in this cohort. We then performed variant-level and gene-level association tests to identify risk factors across the allele frequency spectrum. Finally, we performed additional collapsing analyses in 11,933 individuals of African, East Asian or South Asian genetic ancestry. Using these results, we implemented a pan-ancestry analysis of 281,104 UKB participants. Altogether, this study comprehensively examines the contribution of rare protein-coding variation to the genetic architecture of complex human diseases and quantitative traits.

[1]Centre for Genomics Research, Discovery Sciences, BioPharmaceuticals R&D, AstraZeneca, Waltham, MA, USA. [2]Centre for Genomics Research, Discovery Sciences, BioPharmaceuticals R&D, AstraZeneca, Cambridge, UK. [3]Translational Science and Experimental Medicine, Research and Early Development, Respiratory and Immunology, BioPharmaceuticals R&D, AstraZeneca, Gothenburg, Sweden. [4]Precision Medicine & Biosamples, Oncology R&D, AstraZeneca, Cambridge, UK. [5]Translational Science and Experimental Medicine, Research and Early Development, Respiratory and Immunology, BioPharmaceuticals R&D, AstraZeneca, Cambridge, UK. [6]Department of Medicine, University of Melbourne, Austin Health, Melbourne, Victoria, Australia. [7]Epilepsy Research Centre, University of Melbourne, Austin Health, Melbourne, Victoria, Australia. [20]These authors contributed equally: Quanli Wang, Ryan S. Dhindsa, Keren Carss. *A list of authors and their affiliations appears at the end of the paper. ✉e-mail: slav.petrovski@astrazeneca.com

## Cohort characteristics

We processed 998 TB of raw exome sequence data from 302,355 UKB participants through a cloud-based bioinformatics pipeline (Methods). Through stringent quality control, we removed samples with low sequencing quality and from closely related individuals (Methods). To harmonize variable categorization modes, scaling, and follow-up responses inherent to the phenotype data, we developed PEACOK, a modification of the PHESANT package[24] (Methods).

We considered 17,361 binary traits and 1,419 quantitative traits, which we categorized into 22 ICD-10-based chapters (Extended Data Fig. 1a, b, Supplementary Table 1). We also computed the union of cases across similar phenotypes, resulting in 4,911 union phenotypes (Methods; Supplementary Table 1). The median number of European cases per binary union phenotype was 191 and the median number of individuals tested for quantitative traits was 13,782 (Extended Data Fig. 1c, d). The median number of binary union traits was 25 (Extended Data Fig. 1e).

Approximately 95% of the sequenced UKB participants are of European ancestry (Extended Data Fig. 1f). This affects health-care equity, as the resolution to evaluate variants across the allele frequency spectrum is proportional to the number of sequenced individuals in a population. For example, individuals from non-European ancestries showed a substantially higher number of rare (minor allele frequency (MAF) < 0.005%), non-synonymous variants in Online Mendelian Inheritance in Man (OMIM) disease-associated genes (Extended Data Fig. 1g). This demonstrates a reduced resolution to accurately estimate lower variant frequencies in non-European populations, as previously observed[25].

## Identifying protein-truncating variants

Protein-truncating variants (PTVs), which often inactivate proteins, provide direct insight into human biology and disease mechanisms[26,27]. Identifying PTVs that are protective against disease can also offer direct human validation of potential therapeutic targets[5,28]. Among 287,917 participants of any ancestry, we observed that 96% of 18,762 studied genes had at least one heterozygous PTV carrier, 46% had at least one compound heterozygous or homozygous/hemizygous PTV carrier, and 20% had at least one homozygous/hemizygous PTV carrier (Fig. 1a). Only 884 genes (4.7%) had PTVs with a MAF > 0.5% (Fig. 1a), illustrating the power of exome sequencing to detect this important form of variation. Although some have been implicated in human diseases, most common PTVs occur in genes that are less relevant to disease, such as olfactory receptor genes[29]. Focusing on rarer PTVs (MAF < 1%), we observed that 95% of genes had at least one heterozygous PTV carrier, 42% had at least one compound heterozygous or homozygous/hemizygous PTV carrier, and only 15% had at least one homozygous/hemizygous PTV carrier (Extended Data Fig. 2a).

## Variant-level associations

Exome sequencing enables association tests between phenotypes and protein-coding variants across the allele frequency spectrum. We performed a variant-level exome-wide association study (ExWAS) to test for associations between all 18,780 phenotypes and 2,108,983 variants observed in at least six participants of European ancestry (that is, MAF > 0.001%). We used three genetic models (Methods), equating to 118.8 billion tests. We used a two-sided Fisher's exact test for binary traits and linear regression for quantitative traits. Using a $P$ value threshold of $P \le 2 \times 10^{-9}$ (Methods) and excluding the MHC region (chromosome 6: 25–35 Mb), we identified 5,193 significant genotype–phenotype associations for binary traits and 41,754 associations for quantitative traits (Supplementary Table 2, 3).

Many of the significant ExWAS signals arose from rare variants (MAF < 0.5%) (Fig. 1b). The rarest significant variant was a frameshift

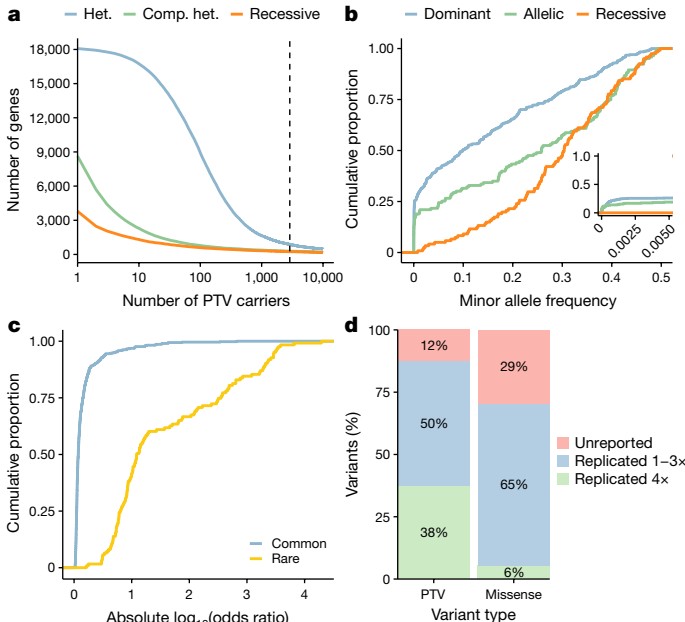

**Fig. 1 | Summary of variant-level exome-wide association study results.**
**a**, The number of genes ($y$ axis) with at least the number of PTV carriers ($x$ axis) in 287,917 UKB participants of any ancestry. The dashed line corresponds to the minimum number of carriers typically required to detect individual PTVs with a MAF > 0.5%, that is, 2,873 carriers. Colours represent heterozygous (het.), putative compound heterozygous (comp. het.) and homozygous/hemizygous carriers (recessive). **b**, The MAF distribution of 632 genome-wide significant ExWAS variants associated with binary traits. The inset plot represents the same data limited to variants with MAF < 0.5%. **c**, The distribution of effect sizes for 509 common versus 123 rare (MAF < 0.5%) significant ExWAS variants. The plots in **b** and **c** include variants with the largest effect sizes achieved per gene. **d**, Percentage of ExWAS study-wide significant PTVs ($n$ = 24) and missense variants ($n$ = 326) that reflect known or novel gene–phenotype relationships. Variants capturing known gene–phenotype relationships were partitioned into those validated in (1) at least one but not all, or (2) all four publicly available databases: FinnGen release r5, OMIM, the GWAS Catalog (including GWAS Catalog variants within a 50-kb flanking sequence either side of the index variant), and the ClinVar pathogenic/likely pathogenic variant collection.

variant in haemoglobin subunit-β (*HBB*) associated with thalassaemia (cohort MAF of 0.0013%) (Supplementary Table 3). In the dominant model, rare variants accounted for 26% of statistically significant associations. Furthermore, 21% (227 of 1,088) of binary trait associations and 12% (1,330 of 10,770) of quantitative trait associations identified using the recessive model were not detected using the dominant model. Associations with more common variants have previously been published[9,12].

The effect sizes of significant rare variant associations were substantially higher than those of common variants (Wilcoxon $P = 1.1 \times 10^{-57}$) (Fig. 1c). While some significant variants are probably in linkage with nearby causal variants, associated PTVs and missense variants often represent the causal variant themselves[26]. Notably, associations for 13% (3 of 24) and 29% (96 of 326) of the significant PTVs and missense variants, respectively, have not been reported in FinnGen release 5, OMIM, ClinVar or the GWAS catalogue[30–32] (Fig. 1d, Supplementary Table 4, 5).

We explored how often significant variant-level associations between different variants in the same gene have opposing directions of effect on a phenotype. Among quantitative trait associations with at least five significant non-synonymous variants (MAF < 0.1%) in a particular gene, at least 80% of variants had the same direction of effect (Extended Data Fig. 2b). This is in contrast to disease-associated non-coding variants, which can variably affect the direction of gene expression[33].

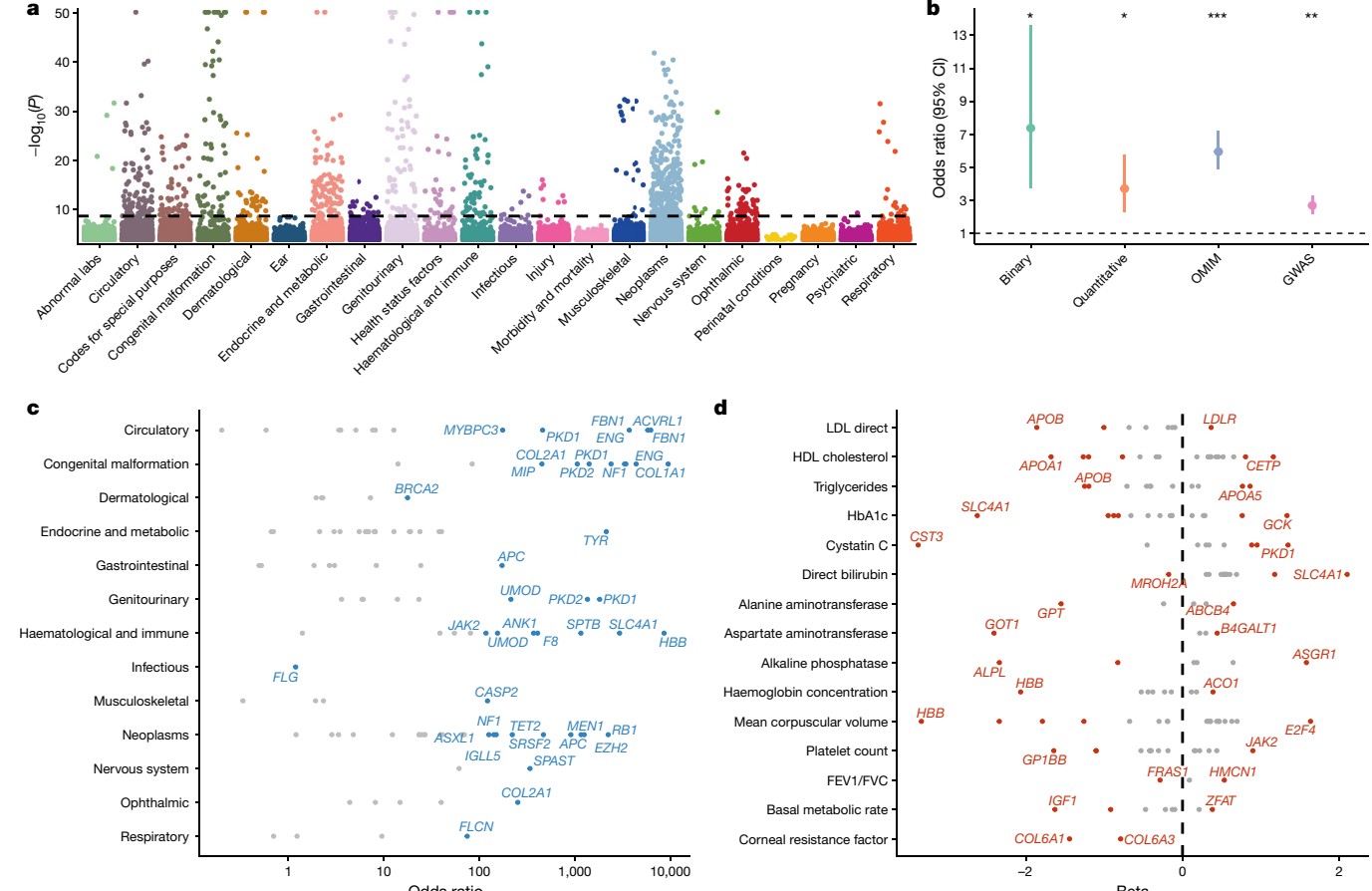

**Fig. 2 | Summary of gene-level collapsing analysis results.**
**a**, Gene–phenotype associations for binary traits. For gene–phenotype associations that appear in multiple collapsing models, we display only the association with the strongest effect size. The dashed line represents the genome-wide significant $P$ value threshold ($2 \times 10^{-9}$). The $y$ axis is capped at $-\log_{10}(P) = 50$ and only associations with $P < 10^{-5}$ were plotted ($n = 94,208$).
**b**, Enrichment of FDA-approved drug targets[6,46] among significant binary traits, quantitative traits, OMIM genes and GWAS signals. $P$ values were generated via two-sided Fisher's exact test (\*$P < 10^{-5}$, \*\*$P < 10^{-20}$, \*\*\*$P < 10^{-70}$). Exact statistics: binary odds ratio (OR) = 7.38, 95% CI: 3.71–13.59, $P = 1.5 \times 10^{-7}$; quantitative OR = 3.71, 95% CI: 2.28–5.76, $P = 4.5 \times 10^{-7}$; OMIM OR = 5.95, 95% CI:

4.90–7.23, $P = 1.1 \times 10^{-75}$; GWAS OR = 2.68, 95% CI: 2.12–3.32, $P = 3.6 \times 10^{-23}$). Error bars represent 95% CIs. Contingency tables were created using each of the binary ($n = 195$), quantitative ($n = 395$), OMIM ($n = 3,875$) and GWAS ($n = 10,692$) categories, alongside approved targets from Informa Pharmaprojects ($n = 463$). $P$ values were generated via a two-tailed Fisher's exact test. **c**, Effect sizes for select gene associations per disease area. Genes with the highest OR for a chapter or with OR > 100 are labelled. **d**, Illustration of large effect gene–phenotype associations for select disease-related quantitative traits. FEV1/FVC, forced expiratory volume in 1 s/forced vital capacity ratio; HDL, high-density lipoprotein; LDL, low-density lipoprotein. Dashed line corresponds to a beta of 0.

We compared the results of our Fisher's exact tests to regression-based frameworks. While an exact test is robust for rarer variants, regression methods can incorporate covariates to help to mitigate confounders and are recommended when careful control for confounding cannot be ensured. We performed single-variant association tests across all autosomal variants for 324 Chapter IX binary phenotypes (diseases of the circulatory system; Supplementary Table 29) using SAIGE SPA[12] and REGENIE 2.0.2 (ref. [34]), including sex, age, sequencing batch and ten principal components as covariates (Supplementary Methods). Fisher's exact Phred scores ($-10 \times \log_{10}(P$ values)) were strongly correlated with those from SAIGE SPA (Pearson's $r = 0.95$) and REGENIE 2.0.2 (Pearson's $r = 0.94$). Fisher's exact $P$ value statistics were also more conservative for lower frequency variants (MAF ≤ 1%) (Supplementary Table 6). Correlation was higher for signals with a $P < 1 \times 10^{-8}$ in either Fisher's exact test or SAIGE SPA (Pearson's $r = 0.99$) and Fisher's exact test or REGENIE 2.0.2 (Pearson's $r = 0.99$) (Supplementary Figs. 1, 2, Supplementary Table 6). The median lambda inflation factor $\lambda_{GC}$ for the Fisher's exact test was 1.0006 (range: 0.9675–1.0698) compared with a median $\lambda_{GC}$ of 0.9953 (range: 0.9372–1.0940) for SAIGE SPA and a median $\lambda_{GC}$ of 1.0001 (range: 0.9439–1.0602) for REGENIE 2.0.2 (Supplementary

Table 7). Finally, we found that the Fisher's exact test was the most computationally efficient of the three methods (Supplementary Table 6). In this setting, the Fisher's exact test offered a statistically robust and efficient alternative to regression-based approaches, but required careful quality control, case–control harmonization and ancestry pruning before association testing.

## Rare variant collapsing analyses

We also performed gene-level association tests using collapsing analyses. In this approach, the proportion of cases with a qualifying variant was compared with the proportion of controls with a qualifying variant in each gene[16–22]. We used 12 different sets of qualifying variant filters (models) to test the association between 18,762 genes and 18,780 phenotypes (Methods; Extended Data Table 1), equating to 4.2 billion tests. The models included ten dominant models, one recessive model and one synonymous variant model that served as an empirical negative control (Methods).

Defining a significance threshold posed a challenge due to strong correlation between the 12 models and among the assessed phenotypes.

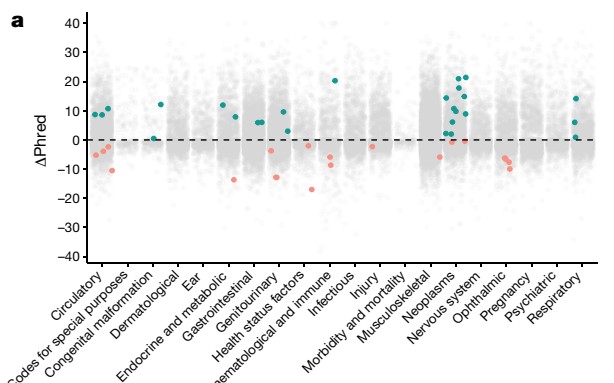
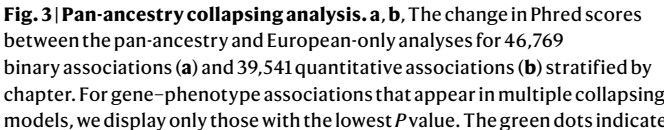
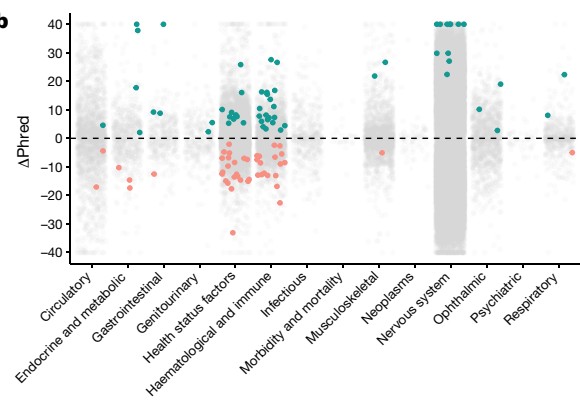

**Fig. 3 | Pan-ancestry collapsing analysis. a, b,** The change in Phred scores between the pan-ancestry and European-only analyses for 46,769 binary associations (**a**) and 39,541 quantitative associations (**b**) stratified by chapter. For gene–phenotype associations that appear in multiple collapsing models, we display only those with the lowest *P* value. The green dots indicate associations that were not significant in the European analysis but were significant in the combined analysis. The orange dots represent associations that were originally significant in the European-only analysis but became not significant in the combined analysis. In both figures, the *y* axis is capped at ΔPhred = 40 (equivalent to a *P* value change of 0.0001).

To avoid false claims, we defined two null distributions: an empirical null distribution using the synonymous collapsing model and an *n*-of-1 permutation-based null distribution. These approaches independently converged on a study-wide significance threshold of $P \leq 2 \times 10^{-9}$ (Methods).

We identified 936 significant gene–phenotype relationships for binary traits and 767 for quantitative traits (Fig. 2a, Extended Data Fig. 3, Supplementary Table 8). These associations were enriched for FDA-approved drugs (binary odds ratio (OR): 7.38 (95% CI: 3.71–13.59), $P = 1.46 \times 10^{-7}$; quantitative OR: 3.71 (95% CI: 2.23–5.74), $P = 7.04 \times 10^{-9}$) (Fig. 2b, Extended Data Fig. 4; Methods) and spanned most disease areas and disease-relevant biomarkers (Fig. 2c, d). Many signals were of large effect, with a median OR of 12.4 for binary traits and a median absolute beta of 0.35 for quantitative traits. We also detected several significant genes with putatively protective PTVs, including *APOB* and *PCSK9* (Supplementary Table 9). The median genomic inflation factor (λ) was 1.002 for binary traits (range: 0.71–1.35) and 1.010 for quantitative traits (range: 0.88–1.37) (Extended Data Fig. 5a). Only 0.76% of the associations from the 191,037 non-recessive collapsing analyses were outside the 0.9–1.1 λ range. Our tests were thus highly robust to systematic bias and other sources of inflation. Collectively, these findings provide biological insight into common diseases and substrates for future therapeutic development opportunities.

Collapsing models focused on PTVs explained 80% of binary and 55% of quantitative associations. Remaining signals emerged from models that included missense variants. While these results confirm the importance of PTVs, they also emphasize the role of other forms of variation in human disease. We found that using the missense tolerance ratio (MTR) to retain missense variants only in constrained genic sub-regions improved the signal-to-noise ratio. Specifically, 15% (133 of 878) of significant relationships detected via the three MTR-informed models were not detected in analogous models that did not incorporate MTR[35]. Moreover, for phenotype associations where both MTR and non-MTR versions of a model achieved significance, effect sizes were significantly higher in the MTR-informed versions (Mann–Whitney test $P = 0.006$; Supplementary Fig. 3). Thus, MTR appears to effectively prioritize putatively functional missense variation in collapsing analyses of complex disease.

Most binary phenotype associations were supported by OMIM or were annotated as pathogenic/likely pathogenic in ClinVar (88.6%), indicating that we robustly captured high-confidence signals (Supplementary Table 10). However, we also identified rare variant associations with phenotypes beyond those reported in OMIM (Supplementary Table 10). For example, 12.1% of the European cohort carried at least one of the 373 distinct filaggrin (*FLG*) PTVs identified. These individuals had a significantly higher risk of well-known associations, including dermatitis ($P = 5.1 \times 10^{-95}$; OR: 1.96 (95% CI: 1.84–2.08)) and asthma ($P = 3.1 \times 10^{-32}$; OR: 1.24 (95% CI: 1.19–1.28))[36], but were also at risk of under-recognized associations, such as melanoma ($P = 4.7 \times 10^{-13}$; OR: 1.21 (95% CI: 1.15–1.27))[37] and basal cell carcinoma ($P = 9.9 \times 10^{-10}$; OR: 1.19 (95% CI: 1.12–1.25))[38]. Concomitant increases in the levels of vitamin D ($P = 2.3 \times 10^{-131}$; β: 0.15 (95% CI: 0.14–0.16))[39] suggest that the increased risk of skin cancer may be attributable to increased sensitivity to ultraviolet B radiation. This interrogation offers one example of how this phenome-wide resource can uncover a wide spectrum of phenotypes associated with rare variation in any protein-coding gene.

Although our pipeline was tuned for detecting germline variants, we identified seven genes that were significantly associated with haematological malignancies, driven by qualifying variants that appeared to be somatic (Supplementary Tables 11, 12, Supplementary Fig. 4). This supports the potential of blood-based sequencing to yield insight into blood cancer genomes via incidentally detected somatic variants[40].

Compared with two smaller UKB PheWAS studies[10,11], we observed a 1.2-fold and 5.6-fold increase, respectively, in statistically significant gene–trait associations using the same first tranche of 50K UKB data, attributable to both the depth of outcomes studied and differences in methodologies (Extended Data Fig. 5b). Increasing the cohort size from 50,000 to the current full dataset led to an 18-fold increase in statistically significant gene–trait associations using our collapsing method (Extended Data Fig. 5c). Incorporating updated phenotypic data from the July 2020 release resulted in a 24-fold increase in significant associations compared with the 50K data (Extended Data Fig. 5c).

Among significant collapsing analysis signals, only 17% (125 of 724) of binary associations and 58% (446 of 767) of quantitative associations were detectable via ExWAS (Supplementary Table 13A). Conversely, most rare PTV ExWAS associations were detected via collapsing analyses, although the rates were lower for rare missense variants (Supplementary Table 13B). Thus, collapsing analyses can identify rare variant associations that are currently undetectable via single-variant-based approaches (Supplementary Table 14).

## Pan-ancestry collapsing analysis

The inclusion of individuals from non-European ancestries in genetic analyses is crucial for health-care equity and genetic discovery[41]. Therefore, we performed additional collapsing analyses for each major non-European ancestral group (that is, South Asian (*n* = 5,714), African (*n* = 4,744) and East Asian (*n* = 1,475)). We limited each PheWAS to binary

traits with at least five cases in the population and quantitative traits with at least five qualifying variants carriers (Supplementary Table 1).

The only study-wide significant ($P \leq 2 \times 10^{-9}$) binary trait association among the non-European populations was between PTVs in *HBB* and thalassaemia in individuals of South Asian ancestry ($P = 2.7 \times 10^{-46}$; OR = 176.4 (95% CI: 84.1–369.7)) (Supplementary Table 15). We next applied the Cochran–Mantel–Haenszel test to combine the results of the binary trait collapsing analysis across all four studied ancestral groups, including the European population (Methods). This pan-ancestry PheWAS identified 26 unique study-wide significant gene–phenotype associations that were not significant in the European analyses (Fig. 3a, Extended Data Fig. 6a, Supplementary Table 16). Conversely, 20 gene–phenotype associations that were significant in the European analyses did not reach the study-wide significance threshold in the pan-ancestry analysis.

We analysed 1,419 quantitative traits in a linear regression model including individuals of all major ancestral groups, including Europeans (Supplementary Table 1). This model included categorical ancestral groups, the top five ancestry principal components, age and sex as covariates (Methods). We identified 59 significant gene–quantitative trait associations that were originally not significant in the European analyses (Fig. 3b, Extended Data Fig. 6b). These included associations between rare variants in *OCA2* and a younger age of wearing glasses ($P = 4.7 \times 10^{-10}$; β: −0.45 (95% CI: −0.60 to −0.31)), *ASGR1* and reduced low-density lipoprotein cholesterol ($P = 1.7 \times 10^{-9}$; β: −0.26 (95% CI: −0.34 to −0.17)), and others (Supplementary Table 17). In addition, 46 unique associations between genes and quantitative traits, originally significant in the European analyses, were not significant in the combined analysis.

## Discussion

We performed a PheWAS using exome sequences of 269,171 UKB participants of European ancestry combined with records of 18,780 phenotypes, followed by a pan-ancestry analysis that incorporated an additional 11,933 UKB participants of African, East Asian and South Asian ancestries. In total, we identified 46,837 variant-level and 1,703 gene-level statistically significant relationships. Many associations were previously known, but others were either new or associated with phenotype expansions. We also found that these associations were significantly enriched for targets of US Food and Drug Administration (FDA)-approved drugs, reinforcing the importance of human genetics in target identification. When followed up with functional investigation to understand biological mechanisms, these results can help to improve the efficiency of pharmaceutical pipelines, contribute towards safety assessments and reveal repositioning opportunities[7,42].

Our variant-level association tests detected rare variant associations that are not frequent enough to be captured by microarray-based studies (that is, as rare as MAF = 0.0012%). Our gene-level collapsing analyses evaluated the aggregate effect of private-to-rare functional variants, 83% of which were not detected in single-variant tests for binary traits. Among gene-level signals for which an individual variant also achieved significance, we found examples where both common and rare risk variants in these genes contributed to disease burden. This is consistent with previous work demonstrating that common and rare PTVs in *FLG* have similar effect sizes for the risk of early asthma[43].

We used a Fisher's exact test framework for our variant-level and gene-level analyses based on previous success with this approach[16–23]. Limitations of the Fisher's test compared with regression-based approaches[12,34,44,45] include an inability to adjust for covariates. On a subset of traits selected for comparisons, we observed that the Phred scores for significant variants from the Fisher's exact test, SAIGE SPA and REGENIE 2.0.2 were nearly perfectly correlated (Pearson's $r = 0.99$). The Fisher's exact test generated more conservative statistics for rare variants and was associated with increased computational efficiency.

Use of the Fisher's exact test requires extremely careful quality control, case–control harmonization and ancestry pruning. In the absence of these measures, it is crucial to correct for such confounders via a regression-based approach. Future work should focus on in-depth benchmarking for these different methods. Regardless of the approach used, it is essential to define an appropriate study-wide significance threshold, which we addressed using *n*-of-1 permutation and an empirical null distribution using a synonymous negative control model.

The predominant representation of European ancestry in human genomics has negative ethical and clinical consequences[25,41]. Smaller sample sizes limited our ability to detect many associations among individual non-European populations. Performing a combined pan-ancestry PheWAS bolstered the association signal for several binary and quantitative traits. Altogether, these results emphasize the need to establish more diverse biobanks.

The UKB has set an excellent standard for linking genomic and phenotypic data and its dynamic nature will facilitate new opportunities for genetic discovery. In future studies, phenotypes may be refined through combining binary, phenotypic and temporal data. The results of this PheWAS are publicly available (http://azphewas.com/), which we anticipate will help to elucidate disease mechanisms, identify phenotypic expansions and enable the development of human genetically validated drugs.

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

**AstraZeneca Genomics Initiative**

**Bastian R. Angermann[3], Ronen Artzi[2,4], Carl Barrett[8], Maria Belvisi[9], Mohammad Bohlooly-Y[10], Oliver Burren[2], Lisa Buvall[11], Benjamin Challis[12], Sophia Cameron-Christie[2], Suzanne Cohen[13], Andrew Davis[14], Regina F. Danielson[15], Brian Dougherty[8], Benjamin Georgi[3], Zara Ghazoui[2], Pernille B. L. Hansen[11], Fengyuan Hu[2], Magda Jeznach[2], Xiao Jiang[2], Chanchal Kumar[12], Zhongwu Lai[8], Glenda Lassi[3], Samuel H. Lewis[2], Bolan Linghu[8], Kieren Lythgow[2], Peter Maccallum[2], Carla Martins[16], Athena Matakidou[2], Erik Michaëlsson[17], Sven Moosmang[12], Sean O'Dell[2], Yoichiro Ohne[18], Joel Okae[2], Amanda O'Neill[2], Dirk S. Paul[2], Anna Reznichenko[12], Michael A Snowden[14], Anna Walentinsson[12], Jorge Zeron[8] & Menelas N. Pangalos[19]**

[8]Translational Medicine, Research and Early Development, Oncology R&D, AstraZeneca, Waltham, MA, USA. [9]Research and Early Development, Respiratory and Immunology, BioPharmaceuticals R&D, AstraZeneca, Cambridge, UK. [10]Translational Genomics, Discovery Biology, Discovery Sciences, BioPharmaceuticals R&D, AstraZeneca, Gothenburg, Sweden. [11]Biosciences CKD, Research and Early Development, Cardiovascular, Renal and Metabolism, BioPharmaceuticals R&D, AstraZeneca, Gothenburg, Sweden. [12]Translational Science & Experimental Medicine, Research and Early Development, Cardiovascular, Renal and Metabolism, BioPharmaceuticals R&D, AstraZeneca, Gothenburg, Sweden. [13]Bioscience Asthma, Research and Early Development, Respiratory & Immunology, Biopharmaceuticals R&D, AstraZeneca, Cambridge, UK. [14]Discovery Sciences, BioPharmaceuticals R&D, AstraZeneca, Gothenburg, Sweden. [15]Research and Early Development, Cardiovascular, Renal and Metabolism, BioPharmaceuticals R&D, AstraZeneca, Gothenburg, Sweden. [16]Oncology Discovery, Early Oncology, Oncology R&D, AstraZeneca, Cambridge, UK. [17]Early Clinical Development, Research and Early Development, Cardiovascular, Renal and Metabolism, BioPharmaceuticals R&D, AstraZeneca, Gothenburg, Sweden. [18]Bioscience Asthma, Research and Early Development, Respiratory & Immunology, Biopharmaceuticals R&D, AstraZeneca, Gaithersburg, MD, USA. [19]Biopharmaceuticals R&D, AstraZeneca, Cambridge, UK.

## Methods

### UKB resource

The UKB is a prospective study of approximately 500,000 participants 40–69 years of age at recruitment. Participants were recruited in the UK between 2006 and 2010 and are continuously followed[47]. The average age at recruitment for sequenced individuals was 56.5 years and 54% of the sequenced cohort comprises those of female sex. Participant data include health records that are periodically updated by the UKB, self-reported survey information, linkage to death and cancer registries, collection of urine and blood biomarkers, imaging data, accelerometer data and various other phenotypic end points[1]. All study participants provided informed consent.

### Phenotypes

We studied two main phenotypic categories: binary and quantitative traits taken from the February 2020 data release that was accessed on 27 March 2020 as part of UKB application 26041. To parse the UKB phenotypic data, we developed a modified version of the PHESANT package, which can be located at https://github.com/astrazeneca-cgr-publications/PEACOK. The adopted parameters are available in Supplementary Methods and have been previously introduced in PHESANT (https://github.com/MRCIEU/PHESANT)[24].

The PEACOK R package implementation focuses on separating phenotype matrix generation from statistical association tests. It also allows statistical tests to be performed separately on different computing environments, such as on a high-performance computing cluster or an AWS Batch environment. This package introduces additional functionalities, including the ability to generate phenotypes for every node from a tree-like UKB data code (for example, an ICD-10 code) and to run logistic regression on a binary phenotype with covariates. Various downstream analysis and summarization were performed using R v3.4.3 https://cran.r-project.org. R libraries data.table (v1.12.8; https://CRAN.R-project.org/package=data.table), MASS (7.3-51.6; https://www.stats.ox.ac.uk/pub/MASS4/), tidyr (1.1.0; https://CRAN.R-project.org/package=tidyr) and dplyr (1.0.0; https://CRAN.R-project.org/package=dplyr) were also used.

In total, 44 UKB paths were represented for the binary traits and 49 for the quantitative traits. For UKB tree fields, such as the ICD-10 hospital admissions (field 41202), we studied each leaf individually and studied each subsequent higher-level groupings up to the ICD-10 root chapter as separate phenotypic entities. Furthermore, for the tree-related fields (fields: 20001, 20002, 40001, 40002, 40006 and 41202), we restricted controls to participants who did not have a positive diagnosis for any phenotype contained within the corresponding chapter to reduce potential contamination due to genetically related diagnoses. A minimum of 30 cases were required for a binary trait to be studied.

In addition to studying UKB algorithmically defined outcomes, we constructed a union phenotype for each ICD-10 phenotype. These union phenotypes are denoted by a 'Union' prefix and the applied mappings are available in Supplementary Table 1.

In total, we studied 17,361 binary and 1,419 quantitative phenotypes. For all binary phenotypes, we matched controls by sex when the percentage of female cases was significantly different (Fisher's exact two-sided $P < 0.05$) from the percentage of available female controls. This included sex-specific traits in which, by design, all controls would be same sex as cases. As a result, 10,531 (60.7%) of the binary phenotypes required down sampling of controls to match the case female percentage (Supplementary Table 1). Finally, to allow for more compartmentalized ICD-10 chapter-based analyses, all 18,780 binary and quantitative trait phenotypes were mapped to a single ICD-10 chapter including manual mapping for the non-ICD-10 phenotypes. Chapter mappings are provided in Supplementary Table 1. It is acknowledged that chapter mapping may have the greatest utility for diagnostic, rather than procedural, ICD-10 codes. For procedural codes, genetic associations could be incorrectly interpreted if chapter mappings are relied on. For example, surgical procedures commonly performed for patients with cancer are categorized within the dermatology chapter. Genetic associations reported for these procedures would be categorized within the dermatology chapter, but the underlying disease process is instead most probably reflective of an oncological aetiology.

We subsequently re-analysed the 300Kv1 cohort using the updated Hospital Episode Statistic (HES) and death registry data as released ad hoc by the UKB on July 2020. Among Data-Field 41270 of primary and secondary inpatient diagnoses that contribute to the Union phenotypes, we found on average a 38.1% increase in the number of cases when comparing the April 2017 refresh to the July 2020 refresh. Throughout this article, we adopt the July 2020 refresh data as the default analysis dataset and refer to this update as the '300Kv2' dataset. The effect on case numbers before and after updating to this release are documented in Supplementary Table 1.

### Sequencing

Whole-exome sequencing data for UKB participants were generated at the Regeneron Genetics Center (RGC) as part of a pre-competitive data generation collaboration between AbbVie, Alnylam Pharmaceuticals, AstraZeneca, Biogen, Bristol-Myers Squibb, Pfizer, Regeneron and Takeda with the UKB[2]. Genomic DNA underwent paired-end 75-bp whole-exome sequencing at Regeneron Pharmaceuticals using the IDT xGen v1 capture kit on the NovaSeq6000 platform. Conversion of sequencing data in BCL format to FASTQ format and the assignments of paired-end sequence reads to samples were based on 10-base barcodes, using bcl2fastq v2.19.0. Exome sequences from 302,355 UKB participants were made available to the Exome Sequencing consortium in December 2019. Initial quality control was performed by Regeneron and included sex discordance, contamination, unresolved duplicate sequences and discordance with microarray genotyping data checks[11].

### AstraZeneca Centre for Genomics Research (CGR) bioinformatics pipeline

The 302,355 UKB exome sequences were processed at AstraZeneca from their unaligned FASTQ state. A custom-built Amazon Web Services (AWS) cloud compute platform running Illumina DRAGEN Bio-IT Platform Germline Pipeline v3.0.7 was used to align the reads to the GRCh38 genome reference and perform single-nucleotide variant (SNV) and insertion and deletion (indel) calling. SNVs and indels were annotated using SnpEFF v4.3[48] against Ensembl Build 38.92[49]. We further annotated all variants with their genome Aggregation Database (gnomAD) MAFs (gnomAD v2.1.1 mapped to GRCh38)[27]. We also annotated missense variants with MTR and REVEL scores[35,50].

### Additional quality control

To complement the quality control performed by Regeneron Pharmaceuticals, we passed the 302,355 sequences through our internal bioinformatics pipeline. In addition to what had already been flagged for quality control, we excluded from our analyses 106 (0.035%) sequences that achieved a VerifyBAMID freemix (contamination) level of more than 4%[51], and an additional five sequences (0.002%) where less than 94.5% of the consensus coding sequence (CCDS release 22) achieved a minimum of tenfold read depth[52].

To mitigate a possible increase of variance estimates due to relatedness, we sought to remove related individuals from our analyses. Using exome sequence-derived genotypes for 43,889 biallelic autosomal SNVs located in coding regions as input to the kinship algorithm included in KING v2.2.3[53], we generated pairwise kinship coefficients for all remaining samples.

We used the ukb_gen_samples_to_remove() function from the R package ukbtools v0.11.3[54] to choose a subset of individuals within which no pair had a kinship coefficient exceeding 0.0884, equivalent of up

to third-degree relatives. For each related pair, this function removes whichever member has the highest number or relatives above the provided threshold, resulting in a maximal set. Through this process, an additional 14,326 (4.74%) sequences were removed from downstream analyses.

After the above quality control steps, there remained 287,917 (95.2%) predominantly unrelated sequences of any genetic ancestry that were available for analyses presented in this work.

## Genetic ancestry

For most of the case–control cohort analyses, we restricted the statistical tests to include a homogeneous European genetic ancestry test cohort. We predicted genetic ancestries from the exome data using peddy v0.4.2 with the ancestry labelled 1,000 Genomes Project as reference. [55]. Of the 287,917 UKB sequences, 18,212 (6.3%) had a Pr(European) ancestry prediction of less than 0.99. Focusing on the remaining 269,706 UKB participants, we further restricted the European ancestry cohort to those within ±4 s.d. across the top four principal component means. This resulted in the exclusion of an additional 535 (0.2%) outlier participants. In total, there were 269,171 predominantly unrelated participants of European ancestry who were included in our European case–control analyses. We also used peddy-derived ancestry predictions to perform case–control PheWAS within non-European populations where there were at least 1,000 exome-sequenced individuals available (see the section 'Collapsing analyses'). Through this step, we identified and used 4,744 (Pr(African) > 0.95), 1,475 (Pr(East Asian) > 0.95) and 5,714 (Pr(South Asian) > 0.95) UKB participants for ancestry-independent collapsing analyses.

## ExWAS analyses

The contribution of rare variants to common disease has, until recently, only been assessed for a subset of complex traits. The gnomAD, which includes exome and genome sequencing data of 141,456 individuals, constitutes the largest publicly available next-generation sequencing resource to date[27]. While this resource has undeniably transformed our ability to interpret rare variants and discover disease-associated genes, it is unsuited to the systematic assessment of the contribution of rare variation to human disease as it lacks linked phenotypic data.

We tested the 2,108,983 variants identified in at least six individuals from the 269,171 predominantly unrelated European ancestry UKB exomes. Variants were required to pass the following quality control criteria: minimum coverage 10X; percent of alternate reads in heterozygous variants ≥ 0.2; binomial test of alternate allele proportion departure from 50% in heterozygous state $P > 1 \times 10^{-6}$; genotype quality score (GQ) ≥ 20; Fisher's strand bias score (FS) ≤ 200 (indels) ≤ 60 (SNVs); mapping quality score (MQ) ≥ 40; quality score (QUAL) ≥ 30; read position rank sum score (RPRS) ≥ −2; mapping quality rank sum score (MQRS) ≥ −8; DRAGEN variant status = PASS; variant site is not missing (that is, less than 10X coverage) in 10% or more of sequences; the variant did not fail any of the aforementioned quality control in 5% or more of sequences; the variant site achieved tenfold coverage in 30% or more of gnomAD exomes, and if the variant was observed in gnomAD exomes, 50% or more of the time those variant calls passed the gnomAD quality control filters (gnomAD exome AC/AC_raw ≥ 50%).

Variant-level $P$ values were generated adopting a Fisher's exact two-sided test. Three distinct genetic models were studied for binary traits: allelic (A versus B allele), dominant (AA + AB versus BB) and recessive (AA versus AB + BB), where A denotes the alternative allele and B denotes the reference allele. For quantitative traits, we adopted a linear regression (correcting for age, sex and age × sex) and replaced the allelic model with a genotypic (AA versus AB versus BB) test. For ExWAS analysis, we used a significance cut-off of $P \leq 2 \times 10^{-9}$. To support the use of this threshold in this study, we performed an $n$-of-1 permutation on the binary and quantitative trait dominant model ExWAS. Only 18 of 38.7 billion permuted tests had $P \leq 2 \times 10^{-9}$, and 58 of 38.7

billion permuted tests had $P$ values less than a more liberal cut-off of $1 \times 10^{-8}$ (Supplementary Tables 18, 19). At this conservative $P \leq 2 \times 10^{-9}$ threshold, the expected number of ExWAS PheWAS false positives is 18 out of the 46,947 observed significant associations.

## Collapsing analyses

To perform collapsing analyses, we aggregate variants within each gene that fit a given set of criteria, identified as qualifying variants[17]. Overall, we performed 11 non-synonymous collapsing analyses, including 10 dominant and one recessive model, plus an additional synonymous variant model as an empirical negative control. In each model, for each gene, the proportion of cases was compared to the proportion of controls among individuals carrying one or more qualifying variants in that gene. The exception is the recessive model, where a participant must have two qualifying alleles, either in homozygous or potential compound heterozygous form. Hemizygous genotypes for the X chromosome were also qualified for the recessive model. The qualifying variant criteria for each collapsing analysis model are in Extended Data Table 1. These models were designed to collectively capture a wide range of genetic architectures. They vary in terms of allele frequency (from private up to a maximum of 5%), predicted consequence (for example, PTV or missense), and REVEL and MTR scores. On the basis of SnpEff annotations, we defined synonymous variants as those annotated as 'synonymous_variant'. We defined PTVs as variants annotated as exon_loss_variant, frameshift_variant, start_lost, stop_gained, stop_lost, splice_acceptor_variant, splice_donor_variant, gene_fusion, bidirectional_gene_fusion, rare_amino_acid_variant, and transcript_ablation. We defined missense as: missense_variant_splice_region_variant, and missense_variant. Non-synonymous variants included: exon_loss_variant, frameshift_variant, start_lost, stop_gained, stop_lost, splice_acceptor_variant, splice_donor_variant, gene_fusion, bidirectional_gene_fusion, rare_amino_acid_variant, transcript_ablation, conservative_inframe_deletion, conservative_inframe_insertion, disruptive_inframe_insertion, disruptive_inframe_deletion, missense_variant_splice_region_variant, missense_variant, and protein_altering_variant.

Collapsing analysis $P$ values were generated by using a Fisher's exact two-sided test. For quantitative traits, we used a linear regression, correcting for age, sex and age × sex.

For all models (Extended Data Table 1), we applied the following quality control filters: minimum coverage 10X; annotation in CCDS transcripts (release 22; approximately 34 Mb); at most 80% alternate reads in homozygous genotypes; percent of alternate reads in heterozygous variants ≥ 0.25 and ≤ 0.8; binomial test of alternate allele proportion departure from 50% in heterozygous state $P > 1 \times 10^{-6}$; GQ ≥ 20; FS ≤ 200 (indels) ≤ 60 (SNVs); MQ ≥ 40; QUAL ≥ 30; read position rank sum score ≥ −2; MQRS ≥ −8; DRAGEN variant status = PASS; the variant site achieved tenfold coverage in ≥ 25% of gnomAD exomes, and if the variant was observed in gnomAD exomes, the variant achieved exome z-score ≥ −2.0 and exome MQ ≥ 30.

To quantify how well a protein-coding gene is represented across all individuals by the exome sequence data, we estimated informativeness statistics for each studied gene on the basis of sequencing coverage across the available exomes (Supplementary Methods, Supplementary Table 24). Moreover, we created dummy phenotypes to correspond to each of the four exome sequence delivery batches to identify and exclude from analyses genes and variants that reflected sequencing batch effects; we provide these as a cautionary list resource for other UKB exome researchers (Supplementary Methods, Supplementary Tables 25–27).

For the pan-ancestry analysis, a Cochran–Mantel–Haenszel test was performed to generate a combined $2 \times 2 \times N$ stratified $P$ value, with $N$ representing up to all four genetic ancestry groups. This was performed for 4,836 binary phenotypes where one of the three non-European ancestries had five or more cases and for all quantitative traits. For the

quantitative traits, we used a linear regression model that included the following covariates: categorical ancestry (European, African, East Asian or South Asian), the top five ancestry principal components, age and sex.

### Compute processing times

Our end-to-end (CRAM → FASTQ → BAM → VCF) processing of the 302,355 UKB exomes was achieved at an average rate of 1,600 exomes per hour, consuming a total of 52,000 hours of CPU time running on Linux servers with FPGA acceleration.

Regarding our collapsing PheWAS analyses, construction of the full set of genotype and phenotype matrices took 13,000 and 30 CPU hours to compile, respectively. The preprocessing steps such as rebalancing sex-specific case–control ratios are incorporated in the phenotype matrix construction time. Subsequently, the approximately 4.5 billion collapsing analysis statistical tests were calculated in 19,000 CPU hours. In wall-clock hours, this took 30 h to generate all the collapsing and phenotype matrices. Once the intermediate files were ready, the roughly 4.5 billion collapsing statistical tests took 8 h to complete.

Regarding our variant-level ExWAS, upon construction of our variant matrices, which took 2,500 CPU hours to compile, all 108 billion statistical tests were calculated in 855,000 CPU hours. In wall-clock hours, this took 37 h to generate the variant matrices. Once these intermediate files were ready, the approximately 108 billion ExWAS statistical tests took 27 h for binary traits and 11 h for quantitative traits.

### Defining the study-wide significant cut-offs for collapsing analyses

Bonferroni correction for multiple testing was inappropriate to use in this study given the high degree of correlation among the studied phenotypes and the level of similarity among the multiple collapsing models. Thus, we took two approaches to define more appropriate study-wide significance thresholds for the gene-based collapsing PheWAS.

We used a synonymous collapsing analysis model as an empirical negative control. Here it is expected that synonymous variants will generally not significantly contribute to disease risk and could thus act as a useful empirical negative control for study-wide $P$ value thresholding. Across the 17,361 studied binary phenotypes and 18,762 studied genes, we observed a distribution of 325,727,082 Fisher's exact test statistics corresponding to the synonymous collapsing model. At the tail of this distribution for binary traits, we identified two genuine relationships: *IGLL5* synonymous variants enriched among 'Union#C911#C91.1 chronic lymphocytic leukaemia' ($P = 2.5 \times 10^{-11}$) and its parent node 'Union#C91#C91 lymphoid leukaemia' ($P = 1.2 \times 10^{-10}$). Following this, we observed a tail of $P$ values beginning from $P = 2.2 \times 10^{-8}$ (Supplementary Table 20). Similarly, for the 1,419 quantitative phenotypes, we observed a distribution of 26,623,278 Fisher's exact test statistics corresponding to the synonymous collapsing model. At the tail of this distribution, we identified two genuine relationships: *MACROD1* synonymous variants correlating with decreased levels of 'Urate' ($P = 2.8 \times 10^{-30}$)[56] and *ALPL* synonymous variants correlating with decreased levels of 'alkaline phosphatase' ($P = 9.3 \times 10^{-9}$)[57]. Following this, we saw a tail of $P$ values beginning from $P = 5.2 \times 10^{-8}$ (Supplementary Table 20).

With this magnitude of test statistics generated in the PheWAS scale, another proposal for $P$ value thresholding involves $n$-of-1 permutation[58]. In applying this approach, we shuffled the case–control (or quantitative measurement) labels once for every phenotype while maintaining the participant-genotype structure and across all 11 non-synonymous collapsing models for binary traits (3,582,997,902 tests) and quantitative traits (292,856,058 tests). Reviewing the tails of these two $P$ value distributions, the lowest permutation-based $P$ value achieved was 1.9 $\times 10^{-9}$ (binary tests) and $3.2 \times 10^{-9}$ (quantitative tests).

Given the scale and correlations among this dataset, we found that both of these approaches provide suitable alternatives to the Bonferroni

$P$ value threshold, which in this case would be $P < 1.2 \times 10^{-11}$. Prioritizing the results of the permutation-based approach because it captures the data structure across all our models, we define a conservative study-wide significance cut-off of $P \le 2 \times 10^{-9}$ for the non-synonymous collapsing analysis results presented in this paper (Supplementary Tables 20, 21). Under this conservative threshold, no positive associations are expected under the null for collapsing analyses.

Finally, for each of the 225,360 exome-wide collapsing analyses comprising the collapsing PheWAS (12 models × (17,361 + 1,419) studied phenotypes), we calculated the lambda genomic inflation factor ($\lambda$) after excluding genes achieving exome-wide significance $P < 2.6 \times 10^{-6}$ for that phenotype (Supplementary Tables 22, 23).

### Collapsing analysis enrichment for approved drug targets

We tested for the enrichment of drug targets among collapsing analysis associations using five publicly available lists: a custom list ($n = 387$; https://raw.githubusercontent.com/ericminikel/drug_target_lof/master/data/drugbank/drug_gene_match.tsv) that was originally derived from DrugBank[59], and another four lists[6] that were originally derived from the Informa Pharmaprojects database[46]. These four lists included drug targets from their latest stages of clinical trials, labelled as 'Approved' ($n = 2,620$), 'Phase I Clinical Trial' ($n = 3,365$), 'Phase II Clinical Trial' ($n = 5,479$) and 'Phase III Clinical Trial' ($n = 1,233$).

For each gene tested in the collapsing analysis, we only retained the most significantly associated phenotype. Distinct gene–phenotype relationships from the collapsing analysis were partitioned into three categories (significant: $P < 2 \times 10^{-9}$ (binary $n = 82$, quantitative $n = 269$); suggestive: $2 \times 10^{-9} < P < 1 \times 10^{-7}$ (binary $n = 113$, quantitative $n = 126$); or non-significant: $P > 1 \times 10^{-7}$ (binary $n = 18,551$, quantitative $n = 18,351$)). The relationship between drug target status and gene–phenotype significance was assessed using Fisher's exact test for each gene list. Specifically, for each of the five lists, we created a contingency table that included the number of significant collapsing analysis genes that intersected with the list and the number of genes that did not intersect with the list out of the list of genes tested in the PheWAS ($n = 18,762$). This was performed for both binary and quantitative traits. We also performed enrichment testing for OMIM[32] genes and GWAS Catalog[31] significant hits (both last accessed on 14 July 2020). We included the most significant associations per gene for the GWAS analysis.

### Ethics reporting

The protocols for UKB are overseen by The UK Biobank Ethics Advisory Committee (EAC); for more information see https://www.ukbiobank.ac.uk/ethics/ and https://www.ukbiobank.ac.uk/wp-content/uploads/2011/05/EGF20082.pdf.

### Reporting summary

Further information on research design is available in the Nature Research Reporting Summary linked to this paper.

## Data availability

Association statistics generated in this study are publicly available through our AstraZeneca Centre for Genomics Research (CGR) PheWAS Portal (http://azphewas.com/). All whole-exome sequencing data described in this paper are publicly available to registered researchers through the UKB data access protocol. Exomes can be found in the UKB showcase portal: https://biobank.ndph.ox.ac.uk/showcase/label.cgi?id=170. Additional information about registration for access to the data is available at http://www.ukbiobank.ac.uk/register-apply/. Data for this study were obtained under Resource Application Number 26041.

A custom list of drug targets from DrugBank is available: https://raw.githubusercontent.com/ericminikel/drug_target_lof/master/data/drugbank/drug_gene_match.tsv. A Pharmaprojects-based list of

drug targets is available: https://raw.githubusercontent.com/AbbVie-ComputationalGenomics/genetic-evidence-approval/master/data/target_indication.tsv.

We used data from the OMIM (https://www.omim.org)[32], MTR (http://mtr-viewer.mdhs.unimelb.edu.au)[35], REVEL[50], gnomAD (https://gnomad.broadinstitute.org)[27], EBI GWAS Catalog (https://www.ebi.ac.uk/gwas)[31], ClinVar (https://www.ncbi.nlm.nih.gov/clinvar)[30] and FinnGen release r5 (https://www.finngen.fi/en).

## Code availability

PheWAS and ExWAS association tests were performed using a custom framework, PEACOK (PEACOK 1.0.7), which is an extension and enhancement of PHESANT. PEACOK 1.0.7 is available on GitHub: https://github.com/astrazeneca-cgr-publications/PEACOK/.

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

**Acknowledgements** We thank the participants and investigators in the UKB study who made this work possible (Resource Application Number 26041); the UKB Exome Sequencing Consortium (UKB-ESC) members AbbVie, Alnylam Pharmaceuticals, AstraZeneca, Biogen, Bristol-Myers Squibb, Pfizer, Regeneron and Takeda for funding the generation of the data and Regeneron Genetics Center for completing the sequencing and initial quality control of the exome sequencing data; the AstraZeneca Centre for Genomics Research Analytics and Informatics team for processing and analysis of sequencing data; and M. Hurles and D. Balding for feedback on this manuscript. We acknowledge the participants and investigators of the FinnGen study.

**Author contributions** Q.W., R.S.D. and S.P. designed the study. Q.W., R.S.D., K.C., A.R.H., A.N., I.T., D.V., M.H., S.M., K.R.S. and S.P. performed analyses and statistical interpretation. Q.W., S.V.V.D. and S.W. did the bioinformatics processing. I.T. performed benchmarking with support from Q.W. Q.W., K.C., K.R.S. and S.W. scoped and lead the PheWAS portal development. R.M., A.P., C.H. and S.P. contributed to the organization of the project. Q.W., R.S.D., K.C., A.R.H., A.N., I.T. and S.P. wrote the manuscript. Q.W., R.S.D., K.C., A.R.H., A.N., I.T., D.V., S.V.V.D., A.M., D.M., M.H., S.M., H.O., S.W., K.R.S., R.M., A.P., C.H. and S.P. reviewed the manuscript.

**Competing interests** Q.W., R.S.D., K.C., A.R.H., A.N., I.T., D.V., S.V.V.D., A.M., D.M., M.H., S.M., H.O., S.W., K.R.S., R.M., A.P., C.H. and S.P are current employees and/or stockholders of AstraZeneca.

**Additional information**
**Correspondence and requests for materials** should be addressed to Slavé Petrovski.

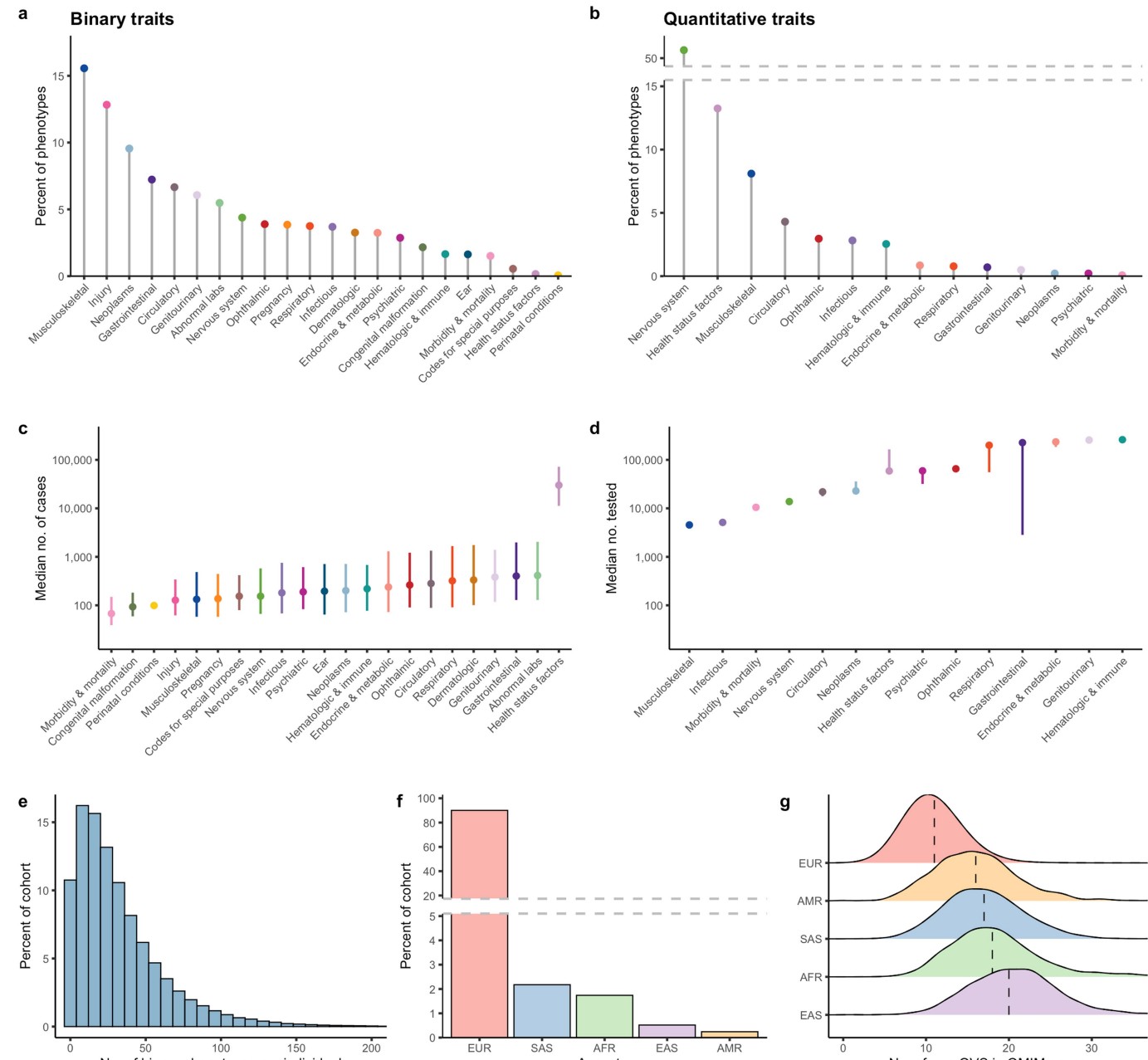

**Extended Data Fig. 1 | Phenotypic and demographic diversity of the sequenced UK Biobank cohort. a**, The percentage of binary union traits assessed in the cohort per disease chapter. **b**, The percentage of quantitative traits assessed in the cohort per chapter. **c**, The median number of cases of European ancestry per binary union phenotype stratified by chapter with interquartile range depicted. The median number of European cases per binary union phenotype was 191 (interquartile range: 72-773). **d**, The median number of participants of European ancestry tested for quantitative traits stratified by chapter with interquartile ranges depicted. The median number of individuals

tested for quantitative traits was 13,782 (interquartile range: 13,780-17,795). **e**, Histogram depicting the number of binary union phenotypes per patient. The *x*-axis was capped at 200 for visual clarity. The median number of binary union traits per European participant was 25 (interquartile range: 12-45) of a possible 4,911. **f**, The distribution of represented genetic ancestries in the sequenced cohort. EUR = European, SAS = South Asian, AFR = African, EAS = East Asian, AMR = American. **g**, The distribution of the number of rare (MAF <0.005%) qualifying variants (QVs) in OMIM-derived Mendelian disease genes per ancestral group. Error bars in (**c**, **d**) represent the interquartile range.

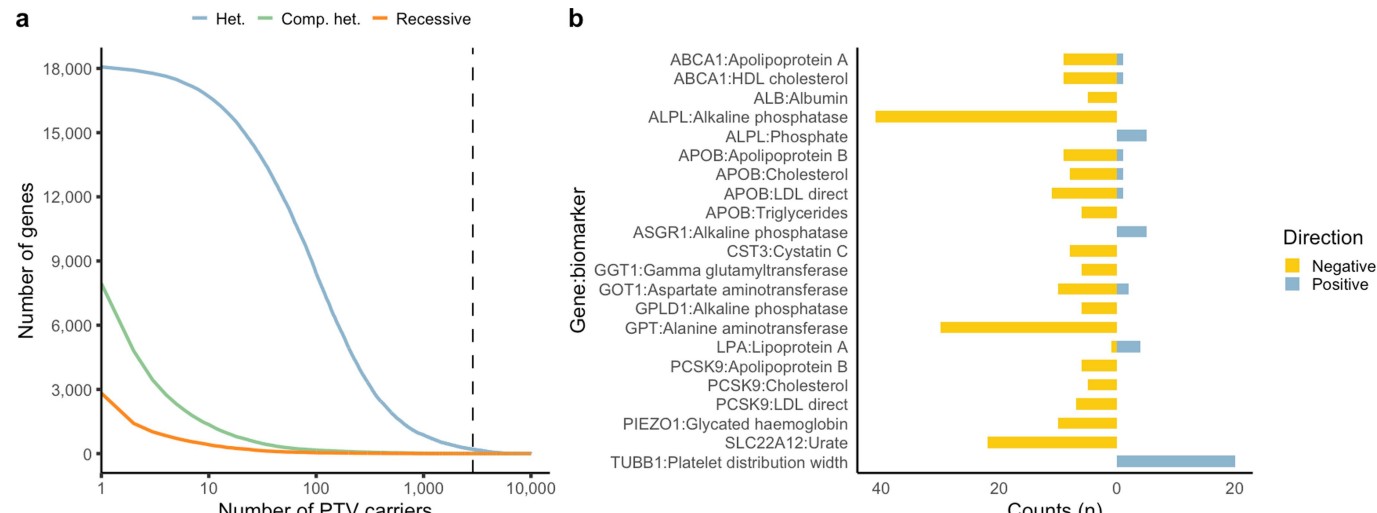

**Extended Data Fig. 2 | Rare PTVs and direction of variant effects. a**, The number of genes (*y*-axis) with at least *N* rare (MAF >0.01) protein-truncating variant (PTV) carriers (*x*-axis) in the cohort. Colours correspond to heterozygous (Het), putative compound heterozygous plus homozygous/hemizygous carriers (comp. het), and exclusively homozygous/hemizygous carriers (recessive). **b**, Distribution of the directions of effect for rare (MAF <0.1%) non-synonymous variant associations with quantitative phenotypes. Only phenotypes with at least five significant non-synonymous variant associations ($P \le 2 \times 10^{-9}$) in a given gene were considered.

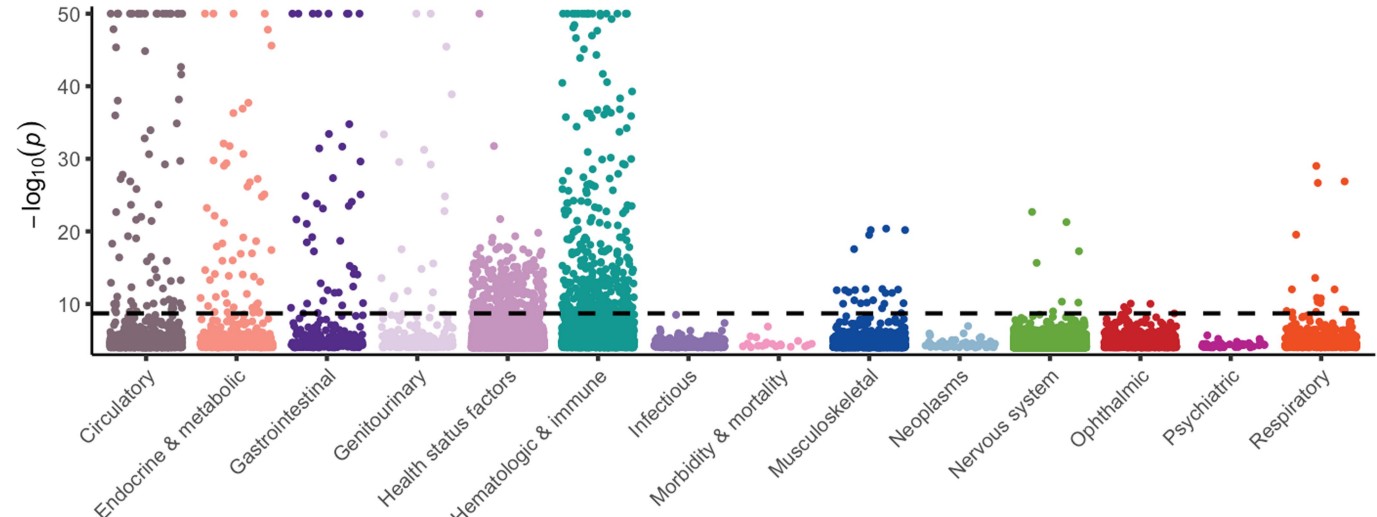

**Extended Data Fig. 3 | Quantitative trait collapsing analysis.** Plot depicting significant gene-phenotype associations for quantitative traits. For gene–phenotype associations that appear in multiple collapsing models, we display only the association with the strongest effect size. The dashed line represents the genome-wide significant p-value threshold ($2 \times 10^{-9}$). The plot is capped at -log10($P$) = 50 and only associations with $P < 10^{-5}$ are included ($n$ = 22,549).

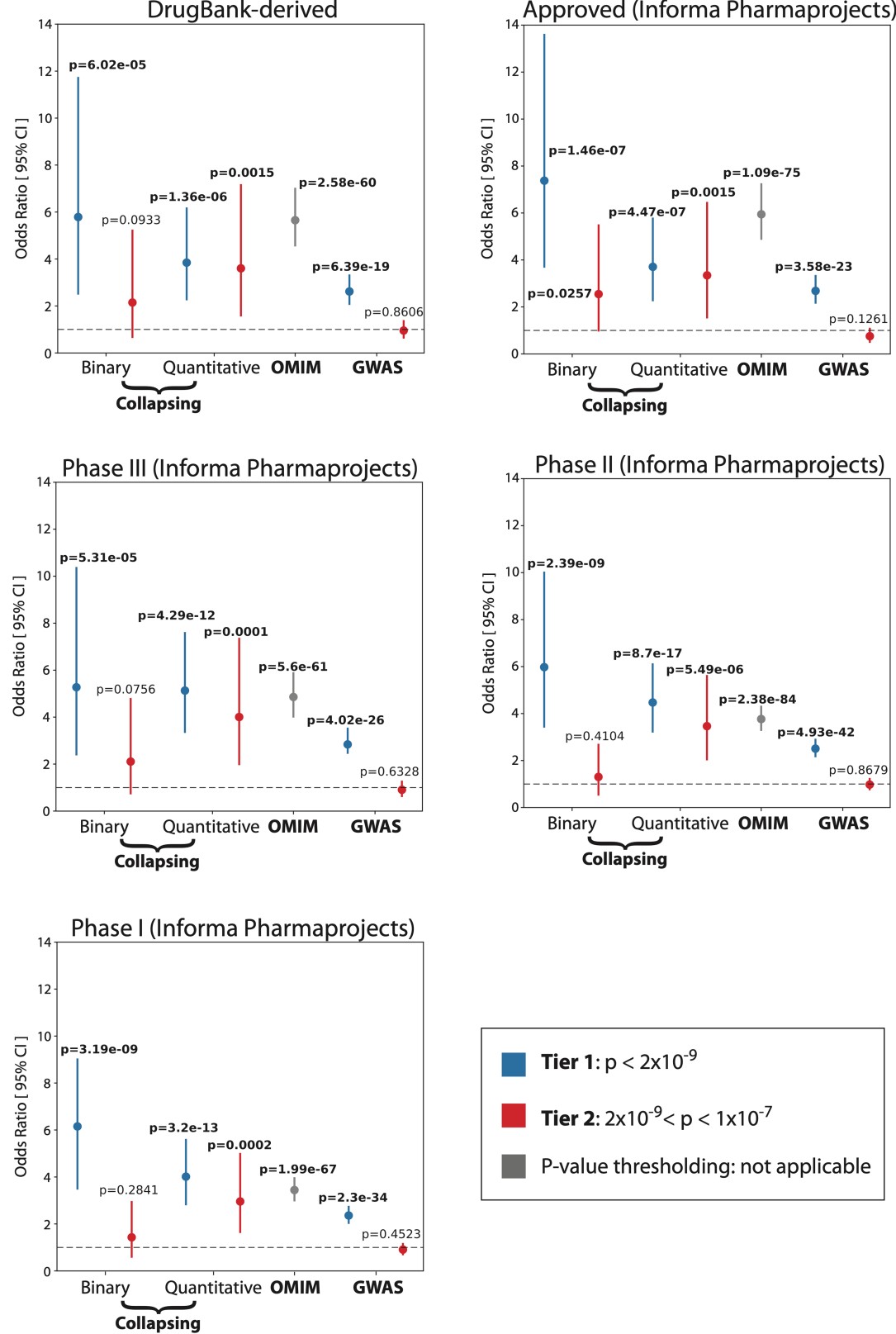

**Extended Data Fig. 4 | Drug target enrichments.** Forest plots demonstrating enrichment of drug targets curated in DrugBank and the Informa Pharmaprojects databases among significant (Tier 1) and nearly significant (Tier 2) binary trait associations, quantitative trait associations, OMIM genes, and GWAS signals. P-values were calculated via Fisher's exact test (two-sided). Error bars represent 95% confidence intervals of the Odds Ratio. The total numbers of genes per category are as follows: DrugBank-derived ($n = 386$); Approved from Informa Pharmaprojects ($n = 463$); Phase III from Informa Pharmaprojects ($n = 474$); Phase II from Informa Pharmaprojects ($n = 1006$); Phase I from Informa Pharmaprojects ($n = 921$); Collapsing – Binary (Tier 1 $n = 82$; Tier 2 $n = 113$); Collapsing - Quantitative (Tier 1 $n = 269$; Tier 2 $n = 126$); OMIM ($n = 3875$); GWAS (Tier 1 $n = 8975$; Tier 2 $n = 1717$).

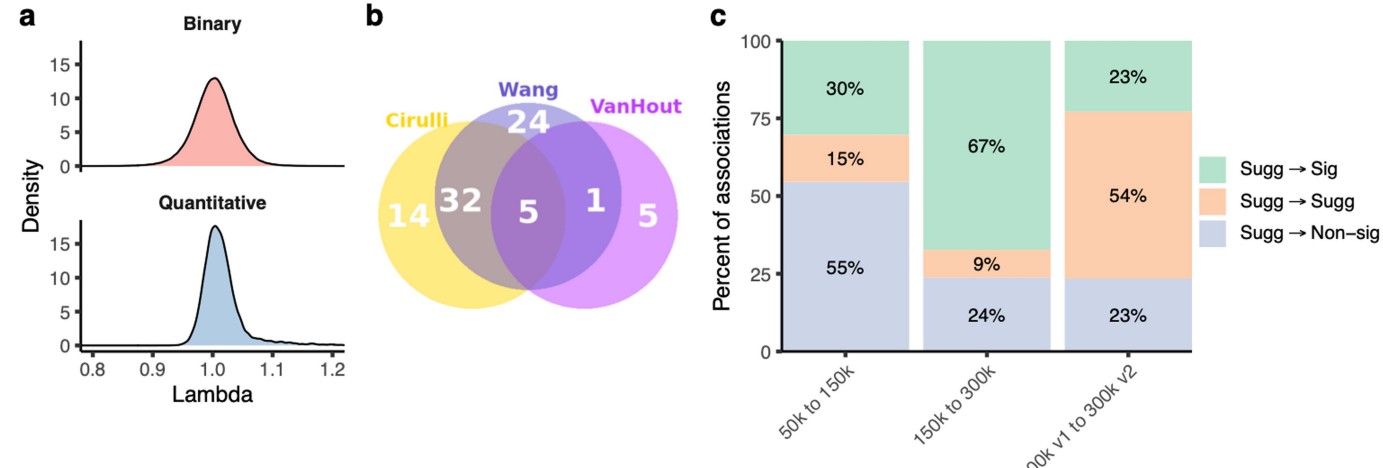

**Extended Data Fig. 5 | Collapsing analysis comparisons. a**, Distribution of lambda (inflation factor) values across all collapsing models for binary and quantitative traits. **b**, Venn diagram for gene-trait associations identified by three studies using the first tranche of 50K UKB. There are 81 distinct significant gene-trait associations ($P < 3.4 \times 10^{-10}$) found among phenotypes that were studied by the three efforts (Supplementary Table 28). **c**, Percentage of suggestive binary gene-phenotype associations that became significant (sig) ($P < 2 \times 10^{-9}$), non-significant (non-sig) ($P > 1 \times 10^{-7}$) or remained suggestive (sugg) ($2 \times 10^{-9} < P < 1 \times 10^{-7}$) with each successive UKB tranche release for binary traits (supplementary methods). 300Kv1 includes phenotypic data released up to April 2017, and 300Kv2 includes additional phenotypic data for the same set of samples released up to July 2020.

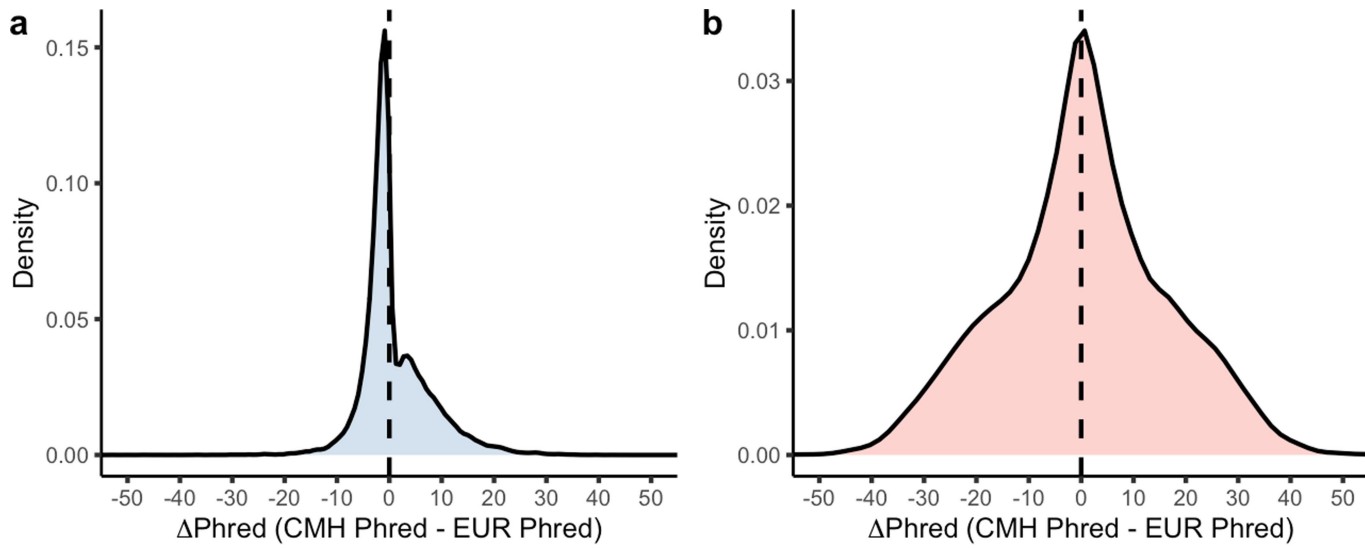

**Extended Data Fig. 6 | Pan-ancestry delta Phred distributions.**
**a**, **b**, Distribution of the change between Phred ((-10*$\log_{10}$[p-values]) scores from the pan-ancestry collapsing analysis and the European-only collapsing analysis for binary traits (**a**) and quantitative traits (**b**). The x-axis in both figures are capped at -50 and +50.

**Extended Data Table 1 | Collapsing analysis models**

| Collapsing model | GnomAD MAF* | UKB MAF | UKB cohort no call or QC fail^ | Variant type | REVEL Cutoff | Missense Tolerance Ratio (MTR) cutoffs |
|---|---|---|---|---|---|---|
| syn (synonymous negative control) | ≤ 0.005% | ≤ 0.05% | ≤ 0.005% | Synonymous | - | - |
| ptv (Protein Truncating) | ≤ 0.1% (popmax) | ≤ 0.1% | ≤ 0.01% | PTV | - | - |
| ptv5pcnt (Protein Truncating, ≤5% MAF) | ≤ 5% (popmax) | ≤ 5% | ≤ 0.5% | PTV | - | - |
| UR (Ultra-rare damaging) | 0% | ≤ 0.005% | ≤ 0.001% | Non-synonymous | ≥ 0.25 | - |
| URmtr (Ultra-rare damaging, MTR informed) | 0% | ≤ 0.005% | ≤ 0.001% | Non-synonymous | ≥ 0.25 | MTR ≤ 25th %ile or intragenic MTR ≤ 50th %ile |
| raredmg (Rare damaging) | ≤ 0.005% | ≤ 0.025% | ≤ 0.005% | Missense | ≥ 0.25 | - |
| raredmgmtr (Rare damaging, MTR informed) | ≤ 0.005% | ≤ 0.025% | ≤ 0.005% | Missense | ≥ 0.25 | MTR ≤ 25th %ile or intragenic MTR ≤ 50th %ile |
| flexdmg (Flexible MAF, damaging non-synonymous) | ≤ 0.1% (popmax) | ≤ 0.1% | ≤ 0.01% | Non-synonymous | ≥ 0.25 | - |
| flexnonsyn (Flexible MAF, all non-synonymous) | ≤ 0.1% (popmax) | ≤ 0.1% | ≤ 0.01% | Non-synonymous | - | - |
| flexnonsynmtr (Flexible MAF, non-synonymous, MTR informed) | ≤ 0.1% (popmax) | ≤ 0.1% | ≤ 0.01% | Non-synonymous | - | MTR ≤ 25th %ile or intragenic MTR ≤ 50th %ile |
| ptvraredmg (PTV or rare damaging models combined) | PTV ≤ 0.1% (popmax) missense ≤ 0.005% and ≤ 0.05% (popmax) | PTV ≤ 0.1% missense ≤ 0.025% | ≤ 0.01% | Non-synonymous | ≥ 0.25 | - |
| rec (Non-synonymous recessive) | ≤ 1% (popmax) ≤ 10 homozygous calls | ≤ 1% | ≤ 0.1% | Non-synonymous | - | - |

"*" reflects the gnomAD global_raw MAF unless otherwise specified. "^" reflects the maximum proportion of UKB exome sequences permitted to either have ≤ 10-fold coverage at variant site or carry a low-confidence variant that did not meet one of the quality-control thresholds applied to collapsing analyses (see Methods).

# Reporting Summary

## Statistics

For all statistical analyses, confirm that the following items are present in the figure legend, table legend, main text, or Methods section.

| n/a | Confirmed | |
|---|---|---|
| ☐ | ☒ | The exact sample size (*n*) for each experimental group/condition, given as a discrete number and unit of measurement |
| ☐ | ☒ | A statement on whether measurements were taken from distinct samples or whether the same sample was measured repeatedly |
| ☐ | ☒ | The statistical test(s) used AND whether they are one- or two-sided<br>*Only common tests should be described solely by name; describe more complex techniques in the Methods section.* |
| ☐ | ☒ | A description of all covariates tested |
| ☐ | ☒ | A description of any assumptions or corrections, such as tests of normality and adjustment for multiple comparisons |
| ☐ | ☒ | A full description of the statistical parameters including central tendency (e.g. means) or other basic estimates (e.g. regression coefficient) AND variation (e.g. standard deviation) or associated estimates of uncertainty (e.g. confidence intervals) |
| ☐ | ☒ | For null hypothesis testing, the test statistic (e.g. *F*, *t*, *r*) with confidence intervals, effect sizes, degrees of freedom and *P* value noted<br>*Give P values as exact values whenever suitable.* |
| ☒ | ☐ | For Bayesian analysis, information on the choice of priors and Markov chain Monte Carlo settings |
| ☒ | ☐ | For hierarchical and complex designs, identification of the appropriate level for tests and full reporting of outcomes |
| ☐ | ☒ | Estimates of effect sizes (e.g. Cohen's *d*, Pearson's *r*), indicating how they were calculated |

*Our web collection on statistics for biologists contains articles on many of the points above.*

## Software and code

Policy information about availability of computer code

| | |
|---|---|
| Data collection | Single-sample processing, on Amazon Web Services (AWS) cloud compute platform.<br>* Conversion of sequencing data in BCL format to FASTQ format and the assignments of paired-end sequence reads to samples based on 10-base barcodes; bcl2fastq v2.19.0 https://support.illumina.com/sequencing/sequencing_software/bcl2fastq-conversion-software.html<br>* read alignment and variant calling performed on Illumina DRAGEN Bio-IT Platform Germline Pipeline v3.0.7 to align the reads to the GRCh38 genome reference and perform small variant SNV and indel calling. SNVs and indels were annotated using SnpEFF v4.3 against Ensembl Build 38.92. We further annotated all variants with their gnomAD minor allele frequencies (gnomAD v2.1.1 mapped to GRCh38).<br>* For ancestry we used PEDDY v0.4.2 with the ancestry labelled 1K Genomes Project reference sequence data for genetic ancestry predictions.<br>* For relatedness we use ukb_gen_samples_to_remove() function from the R package ukbtools v0.11.3. |
| Data analysis | * PheWAS and exWAS association tests were performed using a custom built frame PEACOK (PEACOK 1.0.7), which is an extension and enhancement of PHESANT. PEACOK 1.0.7 can be found: https://github.com/astrazeneca-cgr-publications/PEACOK/versions/1.0.7<br>* Large-scale compute was done using AWS Batch computing environment.<br>* exWAS association tests for Chapter IX binary traits across all autosomes were performed using SAIGE (SAIGE v0.43 https://github.com/weizhouUMICH/SAIGE) and REGENIE (REGENIE v 2.0.2, https://github.com/rgcgithub/regenie).<br>* Using exome sequence-derived genotypes for 43,889 biallelic autosomal SNVs located in coding regions as input to the kinship algorithm included in KING v2.2.3.<br>* Various downstream analysis and summarization were performed using R v3.4.3 https://cran.r-project.org. R library data.table (v1.12.8), MASS (7.3-51.6), tidyr (1.1.0) and dplyr(1.0.0) were also used. |

For manuscripts utilizing custom algorithms or software that are central to the research but not yet described in published literature, software must be made available to editors and reviewers. We strongly encourage code deposition in a community repository (e.g. GitHub). See the Nature Portfolio guidelines for submitting code & software for further information.

## Data

Policy information about <u>availability of data</u>

All manuscripts must include a <u>data availability statement</u>. This statement should provide the following information, where applicable:
- Accession codes, unique identifiers, or web links for publicly available datasets
- A description of any restrictions on data availability
- For clinical datasets or third party data, please ensure that the statement adheres to our <u>policy</u>

All WES data described in this paper are publicly available to registered researchers through the UKB data-access protocol. Additional information about registration for access to the data are available at http://www.ukbiobank.ac.uk/register-apply/. Data for this study were obtained under Resource Application Number 26041. Furthermore, a web portal to interact with the 200K version of these analyses is now publicly available at www.azphewas.com

# Field-specific reporting

Please select the one below that is the best fit for your research. If you are not sure, read the appropriate sections before making your selection.

☒ Life sciences ☐ Behavioural & social sciences ☐ Ecological, evolutionary & environmental sciences

For a reference copy of the document with all sections, see nature.com/documents/nr-reporting-summary-flat.pdf

# Life sciences study design

All studies must disclose on these points even when the disclosure is negative.

| | |
|---|---|
| Sample size | We used the 302,355 UK Biobank participants for whom exome sequencing were available in December 2019. Further subsetting was applied as described in the manuscript. No sample size calculations for power were performed. |
| Data exclusions | At the sample level, predefined exclusion criteria as detailed in the manuscript were: did not pass sequencing quality control thresholds, and are related. |
| Replication | Identified signals were also compared to existing catalogues and other biobank-based PheWAS (namely Finngen public r5, ClinVar and OMIM) and were highly successful with 88.6% of the binary trait collapsing analysis gene-phenotype examples overlapping with these replication sets. |
| Randomization | This study is observational. Randomization was not applicable to this study. |
| Blinding | This study is observational, using coded de-identified data. Blinding was not applicable to this study. |

# Reporting for specific materials, systems and methods

We require information from authors about some types of materials, experimental systems and methods used in many studies. Here, indicate whether each material, system or method listed is relevant to your study. If you are not sure if a list item applies to your research, read the appropriate section before selecting a response.

### Materials & experimental systems

| n/a | Involved in the study |
|---|---|
| ☒ | ☐ Antibodies |
| ☒ | ☐ Eukaryotic cell lines |
| ☒ | ☐ Palaeontology and archaeology |
| ☒ | ☐ Animals and other organisms |
| ☐ | ☒ Human research participants |
| ☒ | ☐ Clinical data |
| ☒ | ☐ Dual use research of concern |

### Methods

| n/a | Involved in the study |
|---|---|
| ☒ | ☐ ChIP-seq |
| ☒ | ☐ Flow cytometry |
| ☒ | ☐ MRI-based neuroimaging |

## Human research participants

Policy information about <u>studies involving human research participants</u>

| | |
|---|---|
| Population characteristics | Of the 302,355 UK Biobank participants for whom exome sequencing were available in December 2019, the average age at recruitment was 56.5 years, 54% were female, and 95% were of European ancestry. Additional population characteristics of the overall UK Biobank cohort are available to the public at http://www.ukbiobank.ac.uk/ |
| Recruitment | Participants were recruited to the UK Biobank on a voluntary basis. Approx 500K individuals 40-69 years of age in 2006-2010 volunteered. Informed consent was obtained for all participants. It has previously been observed that participants are less |

likely to live in socioeconomically deprived areas than non-participants, and they tend to be healthier than non-participants, which may impact some of the reporting rates in comparison to what could be observed through random sampling from the UK population.
Fry et al (10.1093/aje/kwx246).

Ethics oversight

The protocols for UK Biobank are overseen by The UK Biobank Ethics Advisory Committee (EAC), for more information see https://www.ukbiobank.ac.uk/ethics/ and https://www.ukbiobank.ac.uk/wp-content/uploads/2011/05/EGF20082.pdf

Note that full information on the approval of the study protocol must also be provided in the manuscript.

