## [Peer Review File · Nature]

Manuscript Title: Rare variant contribution to human disease in 281,104 UK Biobank exomes

Editorial Notes:

Reviewer Comments & Author Rebuttals

Reviewer Reports on the Initial Version:

Referee #1 (Remarks to the Author):

This is an important paper with very interesting findings for the genetics of human disease. It is the largest yet rare variant 'protein coding variant' wide association study (or exWAS as the authors describe it) for very many multiple phenotypes. With various other studies/analyses following on from this.

Previous manuscripts described the exome sequencing and analyses of the first 50k of UK BioBank (refs 11,12 and the authors nicely describe the differences in Supp Methods page 8). Now the authors analyse ~250k and show there is much more to find.

While this may well eventually get published in Nature, it is not there yet and has multiple flaws.

Major Comments:

1. There is an enormous amount of multiple testing, inevitable with the study of 17361 binary and 1419 quant phenotypes * protein coding variants. The authors do attempt some control for this e.g. with permutation methods.

2. However REPLICATION in an independent dataset is the gold standard for GWAS, and for its friend here protein coding-exWAS. This must be done for at least some of the results.

- There are already 500+k exomes sequenced from UK Biobank (being presented for a few traits at ASHG 2020 last month). Whilst I do not know the deal for who writes which paper from which company for UKBB sequencing, it should be possible to look at doing some limited replication in the next 250k!
- There are other cohorts with sequencing e.g. DeCode.
- There are other cohorts with chip data and imputation e.g. Finngen, MVP that might enable testing of low freq variants (TOPMED will impute to MAF 0.01%) and traits.

3. The association test used has not properly controlled for population/ethnic stratification. Yes, the authors have been careful to pick a very white European group based on genetics. But association tests MUST include principal components in the regression analyses. Recently groups have been using 20 PCs derived from PCA using common variants, and another 20 PCs derived from PCA just using rare variants. As far as I can see just a Fishers Exact Test was used here.

As a consequence of 2 and 3 there are some spurious looking unbelievable odds ratios in Fig 1c.

4. I would like to see a bigger analysis/table/supp table of 'known truths' for rare variant associations. Some of this is in Table1, Fig 1e.

5. page 6 line 170. Most researchers are not very interested in common frequency PTV as these are all in olfactory genes etc (MacArthur et al Science 2012). Instead much more interested in rare PTV, please provide breakdowns for rare.

Minor

6. I am a bit sad that related individuals have been discarded. There are methods (e.g. bolt-Imm, gcta64, SAIGE/REGENIE) that will use these individuals. But what the authors have done is not incorrect, it has just lost a bit of power. Perhaps they can discuss this? Can they also state 'predominantly unrelated' or some similar words, as they have gone down to 8% relatedness - there will be third cousins etc in their analysis.

7. The various models in Supp Table 5 are quite carefully chosen. More discussion of this in the main paper please.

8. I disagree re MAF >0.5% for microarray technology. this may be true for affymetrix UKBB genotypes but is not true for Illumina. please remove. also p17 lines 427,428

Other comments

- It is also a bit sad that only white people have been used. This gives a cleaner dataset. But does lose a lot of power that cross-ethnicity analysis brings. The authors do discuss this a bit in the Discussion, but this is a very topical subject.

With best wishes and look forward to seeing a revised manuscript, David van Heel.

Referee #2 (Remarks to the Author):

Wang et al provide results from analyses of largest tranche of UK Biobank Exome Sequencing data to date. They provide two primary analyses after a general description of the dataset : variant and gene-level association tests for protein coding variation. This is primarily a descriptive paper providing the top association results from these analyses, but represent an enormous undertaking to perform several billion association tests. There don't seem to be any technical faults with the manuscript and it provides a hugely important resource for research labs unable to process the UKB exome dataset on their own. However, I strongly hold that for publication of this manuscript in any journal, full summary statistics from all traits and genes should be made public. There is no barrier to this - there are no legitimate privacy concerns and the hosting burden is not major (about 20,000 files with no more than 10 columns and 20,000 rows). In the reporting summary that data availability is merely stated as the availability of raw UKB data. This does not provide any ability for independently validating and replicating these analyses. To give examples of other Nature publications - gnomAD hosts terabytes of data to aid the community and the INTERVAL study has made their summary statistics public. Researchers will no doubt want to know if their favorite gene had enough variants to be tested, and the p values for those tests, and the supplementary tables provided here, while useful, do not show the full extent of the analyses. That is my major concern with the manuscript.

Major comments :

- The number of genes with heterozygous and homozygous PTVs reported in the UKB exome dataset seems disparate than some literature numbers. In the UKB main paper on medRxiv (<https://www.medrxiv.org/content/10.1101/2020.11.02.2022232v1>), page 6 includes a discussion of the number of genes with PTVs. In Van Hout et al, with 50k individuals, there are 17,718 and 789 genes with het and hom PTVs, respectively. In the results from 200k individuals, there are 18,011 and 1,492 genes with at least one het and hom PTV. In Wang et al, in 287k individuals, there are 18,011 and 3,752 genes with het and hom PTVs (based on 96% and 20% of 18,762 genes evaluated, lines 168 and 170).

It is striking that the number of genes harboring heterozygous PTVs is remarkably consistent between the 200k and 280k datasets, but Wang et al report over double the number of homozygous PTVs. The sample size is different (~100k more individuals in this dataset) and

ancestry make-up might also be different. However this still shouldn't account for doubling of number of genes with homozygous PTVs in mostly the same data. Can the authors report their own numbers for the European subset and clarify the discrepancy? Many of the variant and sample QC filters are standard practice, therefore it's surprising such a large difference exists. From the methods section, I don't see any clarity as to what would result in *more* hom PTVs per individual in this dataset. While the supplemental methods section laying out differences in the burden testing analyses between UKB papers is interesting, it does not get at this disparity.

- Could the authors provide clarity on processing times and prices for the AstraZeneca CGR Bioinformatics pipeline? Running all associations seems like a gargantuan effort, and would be useful information to know approximate CPU cost and run-time to appreciate this.

Related to this point, could the authors clarify reasoning for processing FASTQs vs starting from a functionally equivalent VCF? Most researchers reading this manuscript will be working off the UKB-provided pVCF or plink files, and it is unclear how this dataset differs and how significant that difference is.

- While the authors provided the percent of associations identified in the rare variant collapsing analysis that were not identified with the ExWAS, I wasn't clear on the inverse relationship. Intuitively I'd expect virtually all associations identified via ExWAS to show up in the collapsing analysis - unless non-functional non-synonymous variant drown the signal. How many ExWAS significant associations were re-identified with the rare variant collapsing analysis?

- The analysis of enrichment of approved drug targets needs a little more clarity. Prior publications showing the enrichment matched for phenotype similarity of the tested trait and the approved drug. Could the authors perform a similar analysis to add detail here? (especially given the github repository for King et al is relatively detailed : <https://github.com/AbbVie-ComputationalGenomics/genetic-evidence-approval>) .

At minimum it would be helpful to contextualize whether success of drug targets from ExWAS/collapsing analyses from exomes is more informative than GWAS or OMIM genes.

- On line 178 the authors point to the methods section when stating "Adopting a p value threshold of 1×10^{-8} ...". In the methods section for the ExWAS analysis on line 605, they simply state "we adopted a significance cutoff of 1×10^{-8} " which hardly provides reasoning for the cutoff using 3 models on 2 million variants and ~20,000 traits. Can the authors add reasoning for the p value cutoff threshold for the ExWAS?

- Even though there were three models in the ExWAS analysis, very little time was spent discussing results from the different models. For example, how many variants showing recessive effects also showed dominant effects?

- I found the statement on line 192 "it is uncommon to observe both negatively and positively associated rare variants for a given gene-phenotype relationship" a bit severe. Breaking the function of a gene seems more probable than increasing its function with a PTV or missense variant. If looking at rare non-coding variation, does this same effect hold? At least one GTEX paper looking at expression outliers enriched for rare variants has shown when calling expression outliers, there are equal number of genes that are over-expressed and under-expressed in individuals attributable to underlying rare variants <https://www.nature.com/articles/nature24267>

- I don't follow why a synonymous model was not implemented for the collapsing analysis for each nonsynonymous allele frequency threshold. Different allele frequencies will have different error modes, and rare variants with higher allele frequencies are likely to include systematic errors that might potentially throw false positives. In SuppTable 5, there is only a synonymous variant model for MAF 0.05%, which is a driver of the p value threshold for the analysis. Why not implement a

0.1% MAF, 5% MAF, 0.025% MAF and 0.005% MAF synonymous model (or at least the first two).

- Could the authors compare rare variant burden collapsing results between the raredmg and raredmgmtr or the or flexnonsyn and flexnonsynmtr models? That is to say, the additional value of the MTR metric for burden test associations. They describe its use but mention nothing in the results on its value, which would be informative for researchers.
- I find it weird to only provide the tail of the synonymous p value distributions and not the full distribution. Allowing full access to summary statistics would better allow evaluation of pvalue distributions and visualization of qq plots per trait (since most of the binary trait analyses are underpowered, I don't think we'd expect a large overall lambda skew anyway).
- Can the authors detail on their penetrance estimates beginning on line 302? They acknowledge the importance of comparable prevalence to accurately calculate penetrance, but is the prevalence of asthma and dermatitis in UKB comparable to population prevalence in the UK? There is a conflation of genetic effect size and accuracy of diagnoses in the argument, and the point made here was not clear to me.

Minor points :

- Line 87 –I think clinical genomics suffices, clinico-genomics isn't' quite a commonplace word.
- On line 339 – demonstrable complementarity just sounds like a euphemism for disparity & I think it's ok to just say that.

Referee #3 (Remarks to the Author):

A. Summary of the key results

In this work, Wang et al. leverage Whole-Exome Sequencing to perform single-point and collapsing rare variant analyses on 269,171 European participants of UK Biobank. They consider a wide variety of primary and derived phenotypes, both binary and quantitative, from the Biobank in a pheWAS setting. They also highlight one particular novel association between rare variants in HMCN1 and a lung function phenotype, where functional evidence from existing studies points towards putative regulatory mechanisms in pulmonary disease.

B. Originality and significance: if not novel, please include reference

The main value added of this article is the size of the cohort being studied. It is the first GWAS-type paper to study such a large number of exomes, and therefore the first to be able to examine rare variants' effects on phenotypes in such detail. It builds upon existing papers [ref 11,12] for the methodology and approach taken.

C. Data & methodology: validity of approach, quality of data, quality of presentation

0. The methods used for alignment, variant calling and quality control are sound, given the extremely large sample sizes. The authors used a reasonable set of hard thresholds to filter low-quality genotypes and positions.

1. The association methodology, which uses Fisher's exact test for binary traits and a linear regression for quantitative traits, is quite simplistic (see D. below). The use of these unadjusted models should be justified extensively, ideally replaced, or complemented using sensitivity analyses.

2. It would be useful to summarise what PHEASANT does in section 2. of the methods. I would also rephrase "we adopted a union mapping approach" into a more factual "we computed the

union of cases across phenotypes" or similar.

3. Section 6. of the methods describes sample filtering based on ethnicity, excluding all non-European samples. While it is understood that this was done to alleviate the risk of spurious associations, (a) these samples could have been included had the authors chosen to adjust for ethnicity using a more complex model, (b) the authors themselves show these samples exhibit novel, specific and disease-relevant variants in Fig. 1g, and (c) excluding non-Europeans in the first paper to use the UKB WES data is not ideal. However, I do acknowledge that the cumulative proportion of non-Europeans does not exceed ~10%(Fig 1.f).

4. The Data Availability statement does not state whether full exWAS and variant collapsing summary statistics will be publicly available. They should. Given the scope of the paper (phenome-wide single-point exWAS and rare variant collapsing tests), and given the fact that it is the first such analysis of the UKB Exomes dataset, results should be disseminated widely, and results should be query-able by researchers, e.g through a searchable online portal. Variants contributing to every associated collapsing signal should be listed along with their single-variant effects. The presentation of association results, currently in the form of Table 1 (which feels more like a supplementary table) and ST6, could be improved (e.g., reporting only the strongest p-value for gene/phenotype pairs, improving phenotype descriptions, etc).

D. Appropriate use of statistics and treatment of uncertainties

0. The authors mention a Fisher's as the main association test for single-point analyses. This approach is historical and does not correct for well-documented sources of inflation/T1E. I understand the authors wished to include very low allele counts, however the overall impact of using an uncorrected method outweighs the single advantage of being able to include very rare variants in single-point testing. The reason variant aggregation tests exist is to bypass the fundamental limitations of single-point methods at very low allele counts. If it is retained (I don't think it should), Fisher's testing should only be used for those very rare variants. For anything not ultra-rare, up-to-date methods adjusting for ethnic and other covariates, as well as random relatedness effects, should be used. The size of the study is not a justification either, as single-point GWAS studies of the full imputed UK Biobank, which are comparable in terms of sample size and variant numbers, have successfully used subtler models. Using a better model would make it unnecessary to remove non-Europeans and unrelated individuals.

1. A linked point is that some of the lambdas (inflation factors) are very low or very high. This suggests that a non-negligible part of the test statistics and resulting P-value distributions may be poorly calibrated. I would suggest using hard lambda thresholds (e.g. 0.95 - 1.05) to exclude these analyses from further consideration.

2. Rare variant testing. Again, it is surprising to read that the authors have used a Fisher's exact test for rare variant association. As above, the same questions about adjustment of spurious effect apply here. Arguably, the most interesting use of WES is the examination of rare variants. The authors use a collapsing test, which makes important hypotheses regarding the architecture of the underlying signals. The authors mention in passing that they frequently observe concordant directions of effect in genes with multiple associated variants, but this and its relationship to the collapsing test should be discussed rigorously. It is regrettable that the authors did not examine different architectures, such as those modelled by SKAT-type methods, or SKAT-O optimal tests. In particular, I would have liked to see a distribution of the rho parameter, which could have confirmed the authors' hypotheses regarding signal architecture. These methods have historically scaled poorly, however implementations now exist for biobank sized datasets and the authors should use them.

3. Significance threshold. The authors are convincing in their calculations, however I would like to see their threshold confirmed by a calculation that would take into account the effective number of traits, variants and analyses through reduction of the respective correlation matrices (phenotype, LD and z-scores). It would be good if the authors added more details about their permutation method. Finally, in this paper the authors report results from two correlated analyses using the same test, single-point and collapsed. Two different thresholds were used instead of a single study-wide reporting threshold. The authors should convincingly justify this, or use a single

threshold that adjusts for the increased reporting burden caused by the collapsing analysis.

E. Conclusions: robustness, validity, reliability

0. The conclusion of the paragraph starting at line 188 is overly general. A qualifying statement should be added to the last sentence, acknowledging that this is observed only when testing non-synonymous exonic variants. Indeed, isn't this conclusion likely to be wrong when studying non-PT and/or non-exonic variants?

1. Conclusions regarding association signals are conditional on the robustness of the methods used, as discussed previously.
2. The conclusions concerning HMCN1 are backed up by evidence. The authors' analysis shows that HMCN1 variants are associated with lung function phenotypes. Separately, they show that increased expression of this gene may be involved in idiopathic pulmonary fibrosis and discuss some mechanistic pathways. However, as they note, this constitutes only the "beginning of an elucidation" and can be construed as weak from a clinical interest point of view, especially since it is the only signal discussed in detail (see H.).

F. Suggested improvements: experiments, data for possible revision

0. Address methodological questions and points raised above

1. Improve the presentation and dissemination of their results, ideally through the implementation of a searchable portal (solutions exist for single-point signals, e.g. pheweb). It is especially important that variants constituent of the collapsing tests are documented.
2. Build upon the discussion of signals, for example through the discussion of positive controls, as well as further signals with translational potential.

G. References: appropriate credit to previous work?

0. The authors could refer more to previous rare variant analyses, in particular with respect to the following points:

- which methods did previous papers use to aggregate the effect of rare variants (WES, WGS, and even imputed GWAS)?
- to what extent did the choice of methods place hypotheses on the type of signal that could be detected, and which type of signal was indeed detected?

This can then spur an interesting discussion on how the current dataset can be used to validate these hypotheses, and the methods required for that.

1. The authors should report, in particular for collapsing tests, whether any of their gene/phenotype association has been previously determined, and if yes in which context (GWAS, previous rare variant, family studies...) and present supporting references.

H. Clarity and context: lucidity of abstract/summary, appropriateness of abstract, introduction and conclusions

0. Presenting data at this scale is challenging, and this article is overall well-written and constructed. However, perhaps as a consequence of the vast scope of analyses performed, it is not entirely clear what the scientific focus of the paper is. Currently it feels neither like a resource paper (data accessibility/browsing/sharing is too limited), nor an in-depth analysis of what Biobank-scale WGS can tell us about the architecture of human disease (discussions on effects of rare variants are too succinct), nor like an examination of the clinical relevance of such analyses (see below).

1. As mentioned in E., the paragraph about HMCN1 feels somewhat ancillary. This is the 400th signal in terms of strength of association. How did the authors choose it for discussion? Some of the top-associated collapsing tests provide good proof of concepts: ALPL, APOC3, CST3, APOB, GPT... If the authors wish to focus the paper on clinically relevant gene/phenotype associations, the paper would benefit from discussion of those, plus any other additional signals of potential

clinical/drug development importance.

Referee #4 (Remarks to the Author):

This manuscript describes a massive analysis linking rare coding variants identified from whole exon sequencing on DNA from 269,171 participants in the UK Biobank to a large series of binary and quantitative traits identified based on clinical and laboratory data. The authors identified a large number of protein truncating variants (PTVs) and missense mutations and analyzed associations with both individual sequence variants and groups of presumed loss of function variants in the same gene. A large number of known associations were found and there were many new associations identified. The authors make a strong case that the approaches described will eventually lead to new insights into the molecular bases of human disease and suggest promising drug targets.

One weakness of the current manuscript is the authors effort to highlight a specific example of the utility of their approach. They chose to focus on the new association they identified between a PTV in the gene encoding hemicentin 1 (HMCN1), a secreted component of the extracellular matrix that has been suggested to contribute to the organization of cell adhesions, and an increase in the ratio of FEV1 to FVC. The authors correctly point out that the FEV1/FVC ratio is often increased in patients with restrictive lung diseases, including idiopathic pulmonary fibrosis (IPF). They include convincing data showing that expression of HMCN 1 is increased in whole lung RNA from patients with IPF and suggest that this finding validates their approach. However, this change is in the opposite direction of the change suggested by their genetic data. The PTV in HMCN1 would presumably lead to a reduction in functional HMCT1, not the increase that is seen in IPF. This rare variant (with presumed reduction in HMCN1 function) is associated with the increase in FEV1/FVC that the authors used to identify a link to IPF. Of course, there are many other potential explanations for increases in the FEV1/FVC ratio other than restrictive lung disease and the fact that HMCN1 expression is actually increased rather than decreased in IPF suggests that the association identified probably had nothing to do with IPF.

It is therefore unclear why the authors chose to evaluate how HMCN1 expression levels might alter the biology that underlies pulmonary fibrosis. The experiments they include also somewhat miss the point of the effects seen on HMCN1 gene expression seen in IPF. Rather than overexpress HMCN1 (the abnormality seen in IPF lungs) the authors show that knockdown of HMCN1 leads to a reduction in TGF β -induced upregulation of alpha smooth muscle in cultured primary lung fibroblasts. They also show that knockdown causes a reduction in baseline collagen 1a1 expression and claim that this knockdown also inhibits the increase in col1a1 induced by TGF β . However, when one takes into account the effects on baseline collagen expression, TGF β seems to induce similar proportionate increases in both control and HMCN1 KD fibroblasts. Again, since if anything, the effects seen are in the opposite direction of what would be required to connect a decrease in functional HMCN1 to an increase in FEV1/FVC, none of these data appear to shed light on how a PVT in the HMCN1 gene would increase the FEV1/FVC ratio.

In summary, this is a very impressive paper that includes identification of many new genetic variants that might contribute to human disease and/or drug response. Unfortunately, the single example the authors chose to evaluate in more detail does not actually strengthen the manuscript and should either be deleted or replaced by a more convincing example, where the direction of effect of the variant and the underlying biology are more congruent.

Author Rebuttals to Initial Comments:

Dear Editor and Reviewers:

*We would like to thank the editorial team and reviewers for their comments and the opportunity to improve our manuscript. In particular, to address the questions around our choice of test statistic, we include results of SAIGE SPA adjusting for sex, age, sequencing batch and the top 10 ancestry principal components for all variants on chr 1 against all Chapter IX phenotypes. The comparisons show that relying on a regression framework for low frequency variants detectable by exome sequencing results in unstable associations when minor allele counts in cases or controls approach zero; however, for signals achieving a $p < 1 \times 10^{-8}$, the correlation of the Phred scores ($-10 * \log_{10}[p\text{-values}]$) between our Fisher's Exact and SAIGE SPA tests was Pearson's $r = 0.9997$, with the Exact test proving to be more conservative than SAIGE among lower frequency ($MAF \leq 0.01$) variants. At this scale of analyses, compute efficiency is another key consideration and our exact tests were ~5-fold more efficient. Originally, we were preparing these learnings as a separate technical report, but now incorporate the key insights in this revised manuscript.*

Since our initial submission we have campaigned for and received the support to provide a public portal (<http://azphewas.com/>) for interacting with the PheWAS statistics. This portal currently includes gene-level (collapsing analysis) data generated from the 200K exomes released to the UK Biobank research community. We will expand the portal to include variant-level association statistics and additional data as updated phenotypes and additional exome data are released by the UK Biobank. We have also removed the HMCN1 proof-of-principle from this work while we continue to unravel the biology of that finding over the coming years.

Below, we address each of the specific editorial and reviewer comments and cite the location of the edits corresponding to the marked-up version of our resubmission.

Referee #1

This is an important paper with very interesting findings for the genetics of human disease. It is the largest yet rare variant 'protein coding variant' wide association study (or exWAS as the authors describe it) for very many multiple phenotypes. With various other studies/analyses following on from this. Previous manuscripts described the exome sequencing and analyses of the first 50k of UK BioBank (refs 11,12 and the authors nicely describe the differences in Supp Methods page 8). Now the authors analyse ~250k and show there is much more to find. While this may well eventually get published in Nature, it is not there yet and has multiple flaws.

Major Comments:

- 1. There is an enormous amount of multiple testing, inevitable with the study of 17361 binary and 1419 quant phenotypes * protein coding variants. The authors do attempt some control for this e.g. with permutation methods.**
- 2. However REPLICATION in an independent dataset is the gold standard for GWAS, and for its friend here protein coding-exWAS. This must be done for at least some of the results.**
 - There are already 500+k exomes sequenced from UK Biobank (being presented for a few traits at ASHG 2020 last month). Whilst I do not know the deal for who writes which paper from which company for UKBB sequencing, it should be possible to look at doing some limited replication in the next 250k!**

- There are other cohorts with sequencing e.g. DeCode.
- There are other cohorts with chip data and imputation e.g. Finngen, MVP that might enable testing of low freq variants (TOPMED will impute to MAF 0.01%) and traits.

We now annotate all non-synonymous variants in the ExWAS that achieved $p < 1 \times 10^{-8}$ with the Finngen release 4 outputs to further flag variants achieving $p < 1 \times 10^{-4}$ among the Finngen release 4 public data for a comparable phenotype (Supplementary Table 2). This supplements the existing ExWAS annotations based on ClinVar and GWAS Catalog reporting.

We observe high positive control / replication rates with 77% of the $p < 1 \times 10^{-8}$ exWAS non-synonymous variants supported directly or via other variants in the same gene by one or more of: 1) OMIM evidence, 2) Pathogenic/Likely Pathogenic in ClinVar, 3) independently cited in GWAS Catalogue, or 4) independently observed in Finngen release 4 (<http://r4.finngen.fi/>) – now summarised in Supplementary Table 2. Unsurprisingly, the proportion of previously linked signals is (30/118) 25% for $MAF < 0.1\%$ and (416/461) 90% for $MAF \geq 0.1\%$ missense variants and (16/18) 89% and (14/18) 78% for PTVs (Supplementary Table 2). We summarise this in revised manuscript (See pg 7, lines 194– 198).

3. The association test used has not properly controlled for population/ethnic stratification. Yes, the authors have been careful to pick a very white European group based on genetics. But association tests MUST include principal components in the regression analyses. Recently groups have been using 20 PCs derived from PCA using common variants, and another 20 PCs derived from PCA just using rare variants. As far as I can see just a Fishers Exact Test was used here. As a consequence of 2 and 3 there are some spurious looking unbelievable odds ratios in Fig 1c.

In this study, adopting an exact test allowed us to robustly assess significance of variants at a frequency as low as $MAF = 0.001\%$. The limitations of exact tests are that covariates cannot be incorporated. Thus, we focused on the pre-association harmonisation to mitigate these confounders while still allowing for robust test statistics in the rare variant range. We do recognise that we didn't sufficiently qualify the choice of the exact test for this exome study. In the revised paper, we now illustrate what happens to the test statistic distribution if you run SAIGE (with sex, age, exome sequencing batch, and ancestry PC covariates). Relying on a regression framework for such low frequency and case-control imbalanced configurations results in unstable associations when minor allele counts in cases or controls are near zero. However, for signals achieving a $p < 1 \times 10^{-8}$, Pearson's correlation coefficient of the Phred scores ($-10 \times \log_{10}[p\text{-values}]$) between our Fisher's Exact and the SAIGE SPA (with covariates) test was 0.9997 (pgs 9-10, 215-265 in Supplementary Methods, Supplementary Table 5, and main text pg 8 lines 206-220). The Exact tests proved to be more conservative than SAIGE for variants with $MAF \leq 0.01$ (Fig S3-4, Supplementary Table 6) and also required ~5-fold reduced CPU time.

Supplementary Figure 2. $-10 \cdot \log_{10}(p\text{-values})$ from SAIGE (with 10 PC covariates) and current Fisher's exact test for variant-trait pairs with $p\text{-value} < 1 \times 10^{-8}$ for both SAIGE and Fisher's exact test.

Moreover, we compared SAIGE results with and without PCs. Pearson's correlation coefficient between Phred values from SAIGE with 10 PCs from Bycroft et al. versus SAIGE with no PCs ranges between 0.8425 to 0.9998 with a median of 0.99. This result reflects the fact that: 1) we have carefully selected samples of European genetic ancestry, and 2) adjusting for PCs might not be necessary once we have adjusted for the kinship matrix in SAIGE. Adding 10 PCs as covariates in this situation is a conservative approach, which increases SAIGE running times by 1.5-fold (total of 2,257 running hours for SAIGE without PCs, versus 3,464 hours for SAIGE with 10 PCs).

We also now point our readers interested in common variant associations to previously published PheWAS results derived from imputed micro-array data (p7, 187-188). In our study, our primary goal was to showcase the contribution among the rarest end of the variant frequency spectrum, both at individual variant (exWAS) and genic (collapsing) levels. We have also modified figure 2b to emphasize our focus on rare variants.

The comment about the unbelievable OR's is well received. We feel those results are precisely why the community should be excited about access to rare variant data available through sequencing studies on such large cohorts. Looking at Table 1, all PTVs that achieve an OR > 500 are unsurprising as they have well-established roles in monogenic disease: the association between HBB and thalassemia (three PTVs achieve OR>500), UMOD and CKD (three PTVs), RP1 and Hereditary retinal dystrophy (1 PTV), and BRCA2 and BRCA1 and Prophylactic surgery for risk-factors related to

malignant neoplasms (1 PTV each). In response to reviewer comments and to enhance interpretation, we now also include the 95%CI's from Supplementary Table 2 into Table 1 in the re-submission.

4. I would like to see a bigger analysis/table/supp table of 'known truths' for rare variant associations. Some of this is in Table1, Fig 1e.

In addition to Table 1, the ClinVar and GWAS Catalog annotations for rare variants can be identified as fields in Supplementary Table 4 and also summarised in Figure 2d. In particular, for rare variants, we feel that ClinVar is one reliable source of positive controls and accordingly we find that (13/18) 72% of our exWAS significant PTVs with MAF<0.1% have been previously reported as Pathogenic/Likely Pathogenic in ClinVar and this statistic increases to (16/18) 89% if assessing known haploinsufficiency mediated disease at gene level (Supplementary Table 4).

5. page 6 line 170. Most researchers are not very interested in common frequency PTV as these are all in olfactory genes etc (MacArthur et al Science 2012). Instead much more interested in rare PTV, please provide breakdowns for rare.

In the revised submission we have incorporated an additional PTV figure focusing on PTV's with a MAF<1% and also in the corresponding text (pg 6, lines 166-171, Supplementary Fig 1). We retain the original plots to provide the unfiltered summary.

Minor

6. I am a bit sad that related individuals have been discarded. There are methods (e.g. bolt-Imm, gcta64, SAIGE/REGENIE) that will use these individuals. But what the authors have done is not incorrect, it has just lost a bit of power. Perhaps they can discuss this? Can they also state 'predominantly unrelated' or some similar words, as they have gone down to 8% relatedness - there will be third cousins etc in their analysis.

We now include discussion about the benefits and limitations of approaches in the new section that compares the alternative test statistics for a subset of signals (pg 8, lines 206-220 and pg 16 lines 403-417). We have made the requested suggested language around the relatedness threshold (e.g., pg 22, line 565).

7. The various models in Supp Table 5 are quite carefully chosen. More discussion of this in the main paper please.

We have expanded on the key variations of these qualifying variant models and their motivations in the updated text (pg 24, lines 617-619).

8. I disagree re MAF >0.5% for microarray technology. this may be true for affymetrix UKBB genotypes but is not true for Illumina. please remove. also p17 lines 427,428

We have addressed this accordingly in the revised manuscript.

Other comments

- It is also a bit sad that only white people have been used. This gives a cleaner dataset. But does lose a lot of power that cross-ethnicity analysis brings. The authors do discuss this a bit in the Discussion, but this is a very topical subject.

*We whole-heartedly agree with R1 on this point. Not only does this issue sacrifice power, but it also exacerbates genomics healthcare inequality. In Fig. 1g, we show that the number of candidate rare qualifying variants in OMIM genes is higher for non-European individuals due to the lack of sufficiently large reference cohorts for non-European ancestries. This observation is in-line with our previous commentary (Petrovski and Goldstein, 2016). **It is crucial that the field generate equivalent sequencing and rich phenotyping data for non-European populations.** We further emphasise this point in our revised manuscript (pg 18, lines 448 – 452).*

Referee #2:

Wang et al provide results from analyses of largest tranche of UK Biobank Exome Sequencing data to date. They provide two primary analyses after a general description of the dataset : variant and gene-level association tests for protein coding variation. This is primarily a descriptive paper providing the top association results from these analyses, but represent an enormous undertaking to perform several billion association tests. There don't seem to be any technical faults with the manuscript and it provides a hugely important resource for research labs unable to process the UKB exome dataset on their own. However, I strongly hold that for publication of this manuscript in any journal, full summary statistics from all traits and genes should be made public. There is no barrier to this - there are no legitimate privacy concerns and the hosting burden is not major (about 20,000 files with no more than 10 columns and 20,000 rows). In the reporting summary that data availability is merely stated as the availability of raw UKB data. This does not provide any ability for independently validating and replicating these analyses. To give examples of other Nature publications - gnomAD hosts terabytes of data to aid the community and the INTERVAL study has made their summary statistics public. Researchers will no doubt want to know if their favorite gene had enough variants to be tested, and the p values for those tests, and the supplementary tables provided here, while useful, do not show the full extent of the analyses. That is my major concern with the manuscript.

We have released a public portal for interacting with gene-level association statistics $p < 0.1$. This portal currently includes data generated from the 200,000 exomes available to the UK Biobank research community. We provide a link in the manuscript (pg 28, <http://azphewas.com/>) and will expand this portal to include variant-level association statistics and additional data as the UK Biobank releases updated phenotypes and additional exome data to the public.

Major comments :

- The number of genes with heterozygous and homozygous PTVs reported in the UKB exome dataset seems disparate than some literature numbers. In the UKB main paper on medRxiv (<https://www.medrxiv.org/content/10.1101/2020.11.02.2022232v1>), page 6 includes a discussion of the number of genes with PTVs. In Van Hout et al, with 50k individuals, there are 17,718 and 789 genes with het and hom PTVs, respectively. In the results from 200k individuals,

there are 18,011 and 1,492 genes with at least one het and hom PTV. In Wang et al, in 287k individuals, there are 18,011 and 3,752 genes with het and hom PTVs (based on 96% and 20% of 18,762 genes evaluated, lines 168 and 170).

It is striking that the number of genes harboring heterozygous PTVs is remarkably consistent between the 200k and 280k datasets, but Wang et al report over double the number of homozygous PTVs. The sample size is different (~100k more individuals in this dataset) and ancestry make-up might also be different. However this still shouldn't account for doubling of number of genes with homozygous PTVs in mostly the same data. Can the authors report their own numbers for the European subset and clarify the discrepancy? Many of the variant and sample QC filters are standard practice, therefore it's surprising such a large difference exists. From the methods section, I don't see any clarity as to what would result in *more* hom PTVs per individual in this dataset. While the supplemental methods section laying out differences in the burden testing analyses between UKB papers is interesting, it does not get at this disparity.

We were equally surprised by the discrepancies. Looking into it further, we identified two key differences beyond our inclusion of all ancestries:

- 1) Unlike our estimates that include PTVs across the entire allelic frequency spectrum, the medrxiv 200K draft reflect estimates after restricting to PTVs with an AAF<1% (see their Table 3). This is why, among 200K individuals, no gene was found with >100 homozygous PTV carriers in the medrxiv paper whereas there are many well recognised common PTVs impacting human traits like LPL, LPA, NOD2 and FLG, to name a few. In recognition of similar query by R1, we now include additional heterozygous and homozygous/hemizygous curves that reflect the data imposing a comparable MAF<1% filter in Supplementary Figure 1 and also in the revised text (pg 6, lines 168-171).*
- 2) For our statistics we included the X chromosome genes and this includes hemizygous males among counts. Philosophically, we remain motivated to include the X chr genes than exclude for such a summary, but emphasise the hemizygosity inclusion throughout the text for clarity.*

• **Could the authors provide clarity on processing times and prices for the AstraZeneca CGR Bioinformatics pipeline? Running all associations seems like a gargantuan effort, and would be useful information to know approximate CPU cost and run-time to appreciate this.**

Related to this point, could the authors clarify reasoning for processing FASTQs vs starting from a functionally equivalent VCF? Most researchers reading this manuscript will be working off the UKB-provided pVCF or plink files, and it is unclear how this dataset differs and how significant that difference is.

We will be submitting our DRAGEN reprocessed vcf files to the UK Biobank to host alongside the 2017 Functional equivalent in the UK Biobank RAP. This will allow other groups who, like us, adopt a more scalable Dragen platform to have comparable derived files as their internal sequences. We agree with the value of a functional equivalent, but also recognise virtues of having alternative options available.

Our end-to-end (CRAM -> FASTQ -> BAM -> VCF) processing of the UK Biobank 300K exomes was achieved at a rate of 1,600 exomes per hour, consuming a total of 52K hours of CPU time running on Linux servers with FPGA acceleration.

Regarding our collapsing PheWAS analyses, construction of the full set of genotype and phenotype matrices took 13K and 30 CPU hours to compile, respectively. Subsequently, all ~4.5 billion statistical tests were calculated in 19K CPU hours. In wall clock hours, this took 30 hours to generate all the collapsing and phenotype matrices. Once the intermediate files were ready, the ~4.5 billion collapsing statistical tests took 8 hours to complete.

Regarding our variant-level exWAS, upon construction of our variant matrices, which took 2.5K CPU hours to compile, all 108 billion statistical tests were calculated in 855K CPU hours. In wall clock hours, this took 37 hours to generate the variant matrices. Once these intermediate files were ready, the ~108 Billion exWAS statistical tests took 27 hours for binary traits and 11 hours for quantitative traits.

• While the authors provided the percent of associations identified in the rare variant collapsing analysis that were not identified with the ExWAS, I wasn't clear on the inverse relationship. Intuitively I'd expect virtually all associations identified via ExWAS to show up in the collapsing analysis - unless non-functional non-synonymous variant drown the signal. How many ExWAS significant associations were re-identified with the rare variant collapsing analysis?

One of the added, but well-known, complexities of comparing the results of exWAS with collapsing analysis is that collapsing analysis will, by design, enrich for causal gene-phenotype associations. On the other hand, exWAS is more often impacted by accompanying structure. For this reason, we limited this comparison to variants with $MAF < 0.1\%$, which makes it more comparable to the most flexible MAF imposed by the non-recessive collapsing models. This comparison showed that the proportion of gene-phenotype study-wide relationships identified in exWAS that were also captured in the collapsing analysis was much higher when we focussed on PTVs (85/93 [91%] for quantitative and 30/36 [83%] for binary traits), compared to missense variants (122/270 [45%] for quantitative and 42/188 [22%] for binary traits). We include these new statistics in the results (see page 15, lines 372-377) and as a supplementary table (Supplementary Table 13B).

• The analysis of enrichment of approved drug targets needs a little more clarity. Prior publications showing the enrichment matched for phenotype similarity of the tested trait and the approved drug. Could the authors perform a similar analysis to add detail here? (especially given the github repository for King et al is relatively detailed : <https://github.com/AbbVie-ComputationalGenomics/genetic-evidence-approval>) .

At minimum it would be helpful to contextualize whether success of drug targets from ExWAS/collapsing analyses from exomes is more informative than GWAS or OMIM genes.

We expanded this assessment (methods lines 678-704) by comparing the results of our binary and quantitative collapsing analyses with the gene lists provided by King et al. and accordingly updated Fig 2f and an extended set of comparisons as Supplementary Figure 6. We also expanded these analyses by providing benchmarking against OMIM genes and the GWAS catalogue as recommended by the reviewer.

- On line 178 the authors point to the methods section when stating “Adopting a p value threshold of 1×10^{-8} ...”. In the methods section for the ExWAS analysis on line 605, they simply state “we adopted a significance cutoff of 1×10^{-8} ” which hardly provides reasoning for the cutoff using 3 models on 2 million variants and ~20,000 traits. Can the authors add reasoning for the p value cutoff threshold for the ExWAS?

We apologise for lack of this detail, which we agree is important for community. In the revised manuscript we now explain our reasoning for the selection. In brief, we performed an n-of-1 permutation on the binary and quantitative trait dominant model ExWAS. The below Table summarises the findings across the ~35.8 billion binary tests and ~2.9 billion quantitative tests with the number of observations per increasing order of p-value magnitude.

n-of-1 Permutation p-value cut-off threshold	Binary ExWAS	Quantitative ExWAS
$P < 1 \times 10^{-10}$	0	0
$P < 1 \times 10^{-9}$	4	2
$P < 1 \times 10^{-8}$	75	13
$P < 1 \times 10^{-7}$	684	155
$P < 1 \times 10^{-6}$	7582	1462
Total tests performed	~35.8 billion	~2.9 billion

Although we had some events below the $p < 1 \times 10^{-8}$ cut-off (i.e., 75 out of 35.8 billion), we felt that given the total number of tests performed, these permutation-based results were consistent with the $p < 1 \times 10^{-8}$ guidance for ExWAS from earlier work by Fadista et al., (2016), a more stringent cut-off than community GWAS cut-off of $p < 5 \times 10^{-8}$. In recognizing that our primary focus is on rare variants and in order to maximize computational efficiency, we calculated the n-of-1 ExWAS permutations on the dominant model. We now provide additional detail in our revised manuscript on this selection (pg 22, lines 595-600). Alongside the above summary table (Supplementary Table 15), we introduce another new supplemental table reporting the 5,000 lowest p-value signals from the n-of-1 ExWAS permutation analysis for both binary and quantitative traits (Supplementary Table 16).

- Even though there were three models in the ExWAS analysis, very little time was spent discussing results from the different models. For example, how many variants showing recessive effects also showed dominant effects?

We observed that of the distinct genotype-phenotype associations that were identified using the recessive genetic model, 23% (304/1,342) for binary traits and 14% (1,707/12,500) for quantitative traits were not detectable using the dominant genetic model. We have added some additional text to highlight the value of using different genetic models to enhance the identification of associations between variants and phenotypes. See page 7, lines 184 – 187.

- I found the statement on line 192 “it is uncommon to observe both negatively and positively associated rare variants for a given gene-phenotype relationship” a bit severe. Breaking the function of a gene seems more probable than increasing its function with a PTV or missense

variant. If looking at rare non-coding variation, does this same effect hold? At least one GTEx paper looking at expression outliers enriched for rare variants has shown when calling expression outliers, there are equal number of genes that are over-expressed and under-expressed in individuals attributable to underlying rare variants <https://www.nature.com/articles/nature24267>

We now clarify this section by emphasizing that these data are restricted to rare (MAF < 0.1%) non-synonymous variants that are significantly associated with a specific phenotype. In addition, we also now acknowledge and cite the GTEx finding that there is greater potential for variants with opposing effects in the noncoding regulatory sequence. See page 8, lines 203 – 205.

- **I don't follow why a synonymous model was not implemented for the collapsing analysis for each nonsynonymous allele frequency threshold. Different allele frequencies will have different error modes, and rare variants with higher allele frequencies are likely to include systematic errors that might potentially throw false positives. In SuppTable 5, there is only a synonymous variant model for MAF 0.05%, which is a driver of the p value threshold for the analysis. Why not implement a 0.1% MAF, 5% MAF, 0.025% MAF and 0.005% MAF synonymous model (or at least the first two).**

There is a significant computational burden associated with performing this for all synonymous frequency thresholds. We felt that the one collapsing run would be a good demonstration of the general utility of this empirical null and we accompanied it with our n-of-one permutation-based results. We comment further on this in the revised text, as we sought to prioritise computational resources to generate the complementary permutation-based null distribution. See pg 26, lines 669 – 671.

- **Could the authors compare rare variant burden collapsing results between the raredmg and raredmgmtr or the flexnonsyn and flexnonsynmtr models? That is to say, the additional value of the MTR metric for burden test associations. They describe it's use but mention nothing in the results on its value, which would be informative for researchers.**

In addition to improving the effect sizes of significant associations, we found that the MTR metric captures association signals that were not study-wide significant in the corresponding 'non-mtr' model. Of the 878 distinct study-wide significant gene-phenotype associations spanning the flexnonsyn, flexnonsynmtr, UR, URmtr, raredmg and raredmgmtr models, 133 (15.1%) were study-wide significant only among the mtr-informed models. We include a line about this in the text (pg 11, lines 268 – 275).

In the case of gene-phenotype relationships that were captured by both the 'mtr' and the 'non-mtr' versions of a model, we consistently observed signal enhancement in mtr-aware versions (Supplementary Figure 1). For quantitative and binary traits, the effect sizes in the 'mtr' versions were significantly higher, which support optimised specificity (Mann-Whitney test $P=0.006$). We include a line about this in the text (pg 11, lines 273 – 275).

- **I find it weird to only provide the tail of the synonymous p value distributions and not the full distribution. Allowing full access to summary statistics would better allow evaluation of pvalue**

distributions and visualization of qq plots per trait (since most of the binary trait analyses are underpowered, I don't think we'd expect a large overall lambda skew anyway).

The synonymous output represents >300 million statistical tests. In our resubmission we now provide two supplemental files. The first lists the synonymous-based p-values extended to those achieving $p < 0.001$ (Supplementary Table 17). This is supplemented by an additional table summarising the number of positive events along increasing p-value thresholds (Supplementary Table 18 – shared below for convenience). Users will also be able to interact with the synonymous model results as part of the portal <http://azphewas.com/>.

Synonymous collapsing output p-value cut-off threshold	Binary ExWAS	Quantitative ExWAS
$P < 1 \times 10^{-10}$	1*	1*
$P < 1 \times 10^{-9}$	2*	1*
$P < 1 \times 10^{-8}$	2*	2*
$P < 1 \times 10^{-7}$	11	3
$P < 1 \times 10^{-6}$	105	30
$P < 1 \times 10^{-5}$	1,062	271
$P < 1 \times 10^{-4}$	10,967	2,417
$P < 1 \times 10^{-3}$	116,366	22,876
Total tests performed	~346.5 million	~28.3million

* Biological precedence (likely true) associations.

Moreover, we also now provide an additional file with the full list of lambdas as a separate supplemental, which we realise we unintentionally omitted in our initial submission Supplementary Table 19.

• Can the authors detail on their penetrance estimates beginning on line 302? They acknowledge the importance of comparable prevalence to accurately calculate penetrance, but is the prevalence of asthma and dermatitis in UKB comparable to population prevalence in the UK? There is a conflation of genetic effect size and accuracy of diagnoses in the argument, and the point made here was not clear to me.

We do observe that the prevalence of asthma and dermatitis in the UKB is comparable to the population prevalence that has been reported in literature. We have revised the text accordingly – see pages 13-14, lines 321 – 332.

Minor points :

• Line 87 –I think clinical genomics suffices, clinico-genomics isn't quite a commonplace word.

We have revised the text accordingly.

• On line 339 – demonstrable complementarity just sounds like a euphemism for disparity ☹ I think it's ok to just say that.

We have revised the text accordingly.

Referee #3:

C. Data & methodology: validity of approach, quality of data, quality of presentation

1. The association methodology, which uses Fisher's exact test for binary traits and a linear regression for quantitative traits, is quite simplistic (see D. below). The use of these unadjusted models should be justified extensively, ideally replaced, or complemented using sensitivity analyses.

This point is similar to the one raised by R1. In brief, we now clarify that the motivation for this study was the exploration and robust study of rare variant associations. This included studying variants as low as 0.001% MAF. For readers interested in variants with a more common MAF (>0.1%), we now direct them to the existing microarray studies on the same cohort that use the standard common variant best practices. As we do not want to simply ignore variants for being too rare, and to support the use of an exact test in these scenarios, we now provide direct comparison of Fisher's exact and SAIGE (with PC covariates) for a subset of phenotypes on chromosome 1 (pgs 9-10, 215-265 in Supplementary Methods, Fig S2-4; and main text pg 8 lines 206-220).

2. It would be useful to summarise what PHEASANT does in section 2. of the methods. I would also rephrase "we adopted a union mapping approach" into a more factual "we computed the union of cases across phenotypes" or similar.

We have expanded Section 2 of methods and revised the text accordingly.

3. Section 6. of the methods describes sample filtering based on ethnicity, excluding all non-European samples. While it is understood that this was done to alleviate the risk of spurious associations, (a) these samples could have been included had the authors chosen to adjust for ethnicity using a more complex model, (b) the authors themselves show these samples exhibit novel, specific and disease-relevant variants in Fig. 1g, and (c) excluding non-Europeans in the first paper to use the UKB WES data is not ideal. However, I do acknowledge that the cumulative proportion of non-Europeans does not exceed ~10%(Fig 1.f).

We whole-heartedly agree on this point. As mentioned in our response to R1, not only does this issue sacrifice power, but it also exacerbates genomics healthcare inequality. In Fig. 1g, we show that the number of candidate rare qualifying variants in OMIM genes is higher for non-European individuals due to the lack of sufficiently large reference cohorts for non-European ancestries. This observation is in-line with our previous commentary (Petrovski and Goldstein, 2016). It is crucial that the field generate sequencing and rich phenotyping data for non-European cohorts. We further emphasise this point in our revised manuscript (pg 18, lines 448 – 452). However, as this reviewer points out, non-European individuals account for <10% of the cohort. Methods such as SAIGE and BOLTLMM, although they allow for inclusion of PCs, have not been used or tested for transethnic analysis. The appropriate approach for such analysis would be to analyse each ethnicity separately and then combine the results in a meta-analysis framework. In the UK Biobank instance, we expect such results to be overwhelmed by the European representation, and therefore, the non-European ancestry

groups will add little information in this context. We feel this is best addressed by the generation of equivalent medical research resources in non-European collections.

4. The Data Availability statement does not state whether full exWAS and variant collapsing summary statistics will be publicly available. They should. Given the scope of the paper (phenome-wide single-point exWAS and rare variant collapsing tests), and given the fact that it is the first such analysis of the UKB Exomes dataset, results should be disseminated widely, and results should be query-able by researchers, e.g through a searchable online portal. Variants contributing to every associated collapsing signal should be listed along with their single-variant effects. The presentation of association results, currently in the form of Table 1 (which feels more like a supplementary table) and ST6, could be improved (e.g., reporting only the strongest p-value for gene/phenotype pairs, improving phenotype descriptions, etc).

With the accompanying data portal, the PheWAS association statistics are browsable. In addition, we have modified Table 1 to report only the strongest p-value for each gene/phenotype pair and included the 95% CI's. Full phenotype descriptions, including field codes and ICD10 codes where applicable were used in this table purposefully to allow the reader to cross reference Table 1 with Supplementary Table 1. We have now added a comment explaining this to the legend of Table 1.

D. Appropriate use of statistics and treatment of uncertainties

0. The authors mention a Fisher's as the main association test for single-point analyses. This approach is historical and does not correct for well-documented sources of inflation/T1E. I understand the authors wished to include very low allele counts, however the overall impact of using an uncorrected method outweighs the single advantage of being able to include very rare variants in single-point testing. The reason variant aggregation tests exist is to bypass the fundamental limitations of single-point methods at very low allele counts. If it is retained (I don't think it should), Fisher's testing should only be used for those very rare variants. For anything not ultra-rare, up-to-date methods adjusting for ethnic and other covariates, as well as random relatedness effects, should be used. The size of the study is not a justification either, as single-point GWAS studies of the full imputed UK Biobank, which are comparable in terms of sample size and variant numbers, have successfully used subtler models. Using a better model would make it unnecessary to remove non-Europeans and unrelated individuals.

The Fisher's exact test has the advantage of being robust for very low numbers of minor allele counts and highly unbalanced cases-control configurations, while also being ~5-fold more compute efficient. Well known limitations of this approach include the inability to adjust for covariates and to include potentially related individuals, which may have reduced case sample sizes for some studied phenotypes. Software such as BOLT-LMM, SAIGE and REGENIE have been used for running PheWAS on large datasets and accommodate for kinship matrix, traditional covariates and improved handling of imbalanced cases-control studies over traditional regression approaches. Although such approaches are attractive and are improving in scalability, they remain less compute efficient and less statistically robust for very rare variants than an exact test. Non-Europeans are still removed when using such software, as they have not been extensively tested in the setting of trans-ethnic

studies (please see “Frequently asked questions” at the SAIGE/SAIGE-GENE wiki). In our study, we control for confounders in our study design by down-sampling controls to correct for sex imbalance, adopting the PC’s to define the homogenous European ancestry test cohort, as well as well-calibrated p-value significance thresholds, and are confident our results demonstrate that we do not suffer from unreasonable Type I error rates.

We have now also performed variant-level tests on chromosome 1 using SAIGE SPA for Chapter IX - Diseases of the circulatory system, which is a subset of 324 binary traits, adjusting for sex, age, sequencing batch and the top 10 ancestry principal components (See pg 8 lines 206-220 and pg 16, lines 403 – 417). As seen in our newly introduced Supplementary Figure 2, for signals achieving a $p < 1 \times 10^{-8}$ by Fisher’s exact and SAIGE, Pearson’s correlation coefficient (r) of the Phred scores ($-10 \times \log_{10}[p\text{-values}]$) between the Exact and regression test was 0.9997, with the Exact test proving to be more conservative than SAIGE for variants with $MAF \leq 0.01$ (Supplementary Figure 4).

1. A linked point is that some of the lambdas (inflation factors) are very low or very high. This suggests that a non-negligible part of the test statistics and resulting P-value distributions may be poorly calibrated. I would suggest using hard lambda thresholds (e.g. 0.95 - 1.05) to exclude these analyses from further consideration.

We now include the lambda’s for all studied phenotypes and models as an additional supplemental file (Supplementary Table 19). This allows readers to query the associated lambda for a given studied phenotype and genetic model. Important to note that only 1.3% of all 208,405 studied binary collapsing analyses landed outside of a respectable 0.90-1.10 lambda range, with nothing exceeding a lambda of 1.35. Given the scale of this study, we consider this an impressive small set of outliers and feel speaks well to the robustness of the implementation of the adopted analytical framework in the setting of rare-variant analyses. We also highlight that the majority of the events outside the 0.90-1.10 lambda range are driven by the recessive model, and excluding the recessive model from the lambda distribution only 0.76% of 191,037 remaining non-recessive model collapsing analyses land outside the 0.90-1.10 lambda range. We discuss this additional consideration in our revised text (pg 13, lines 300 – 302).

2. Rare variant testing. Again, it is surprising to read that the authors have used a Fisher’s exact test for rare variant association. As above, the same questions about adjustment of spurious effect apply here. Arguably, the most interesting use of WES is the examination of rare variants. The authors use a collapsing test, which makes important hypotheses regarding the architecture of the underlying signals. The authors mention in passing that they frequently observe concordant directions of effect in genes with multiple associated variants, but this and its relationship to the collapsing test should be discussed rigorously. It is regrettable that the authors did not examine different architectures, such as those modelled by SKAT-type methods, or SKAT-O optimal tests. In particular, I would have liked to see a distribution of the rho parameter, which could have confirmed the authors’ hypotheses regarding signal architecture. These methods have historically scaled poorly, however implementations now exist for biobank sized datasets and the authors should use them.

With regard to not adjusting for covariates, we refer the reviewer to our response to comment number 0 above. Regarding the use of a burden test, this is one of the most popular and simple methods, adopted also by Van Hout et al and Cirulli et al when analysing the first tranche of the UK Biobank 50K WES. The assumptions of the burden test are: 1) the variants collapsed together are independent and not in LD, 2) they have a function on the trait of interest, and 3) all of them affect the trait in the same direction. SKAT alleviates assumptions (2) and (3), while SKAT-O lets the data decide what allelic architecture fits the trait-gene pair best. SAIGE-GENE implements all 3 tests and while this is outside the scope of our manuscript, comparing results of the 3 approaches on thousands of traits in UK Biobank would be of interest to the research community. Regarding the assumptions of the burden test, assumption (1) is accurate as we used only very rare variants in gene-based analysis, while assumption (2) should be accurate as we focus predominantly on predicted loss of function and missense variants. Regarding assumption (3), we investigated the consistency in the direction of effect of significant rare non-synonymous variants (pg 8, 203-205) and we show that it is uncommon for those within the same gene to have opposing effects (Fig 2e). In the unlikely scenario where this assumption is violated, we would expect a false negative rather than a false positive result and we made the decision a priori that this would be acceptable.

3. Significance threshold. The authors are convincing in their calculations, however I would like to see their threshold confirmed by a calculation that would take into account the effective number of traits, variants and analyses through reduction of the respective correlation matrices (phenotype, LD and z-scores). It would be good if the authors added more details about their permutation method. Finally, in this paper the authors report results from two correlated analyses using the same test, single-point and collapsed. Two different thresholds were used instead of a single study-wide reporting threshold. The authors should convincingly justify this, or use a single threshold that adjusts for the increased reporting burden caused by the collapsing analysis.

We provide further details on our permutation method (pg 25, lines 660 – 671). We also now provide the n-of-one permutation based assessment for ExWAS. The ExWAS and collapsing represent considerably different test statistic distributions and we consider treating them separately to be appropriate.

E. Conclusions: robustness, validity, reliability

0. The conclusion of the paragraph starting at line 188 is overly general. A qualifying statement should be added to the last sentence, acknowledging that this is observed only when testing non-synonymous exonic variants. Indeed, isn't this conclusion likely to be wrong when studying non-PT and/or non-exonic variants?

We now address this according to the reviewer's suggestion – a valid point also raised by R2. See pg 8, 203-205.

1. Conclusions regarding association signals are conditional on the robustness of the methods used, as discussed previously.

In the revised manuscript we emphasise the focus of this work is on the rare variant spectrum as this is where the value of exome sequencing comes from. We addressed the comments about the choice

*of an exact test statistic in earlier responses (and on pg 16, lines 403 – 417). Briefly, our comparison analysis shows that relying on a regression framework for low frequency variants results in unstable associations when minor allele counts in cases or controls approach zero; however, for signals achieving a $p < 1 \times 10^{-8}$, the correlation of the $-10 * \log_{10}(p\text{-values})$ between the Exact and SAIGE regression test was Pearson's $r = 0.9997$, with the Exact test proving to be more conservative than SAIGE for variants with $MAF \leq 0.01$. As cited in earlier responses for R1, the considerably high rate of positive controls (known true rare-variant or gene-level associations) among our outputs are an indicator of robustness of the approaches taken in this study focusing on maximising value of access to the rarer end of the frequency spectrum to complement existing microarray-based results on the same dataset that are readily available to the public (See pg 7, Lines 186 – 188, pg 17 436 – 438).*

2. The conclusions concerning HMCN1 are backed up by evidence. The authors' analysis shows that HMCN1 variants are associated with lung function phenotypes. Separately, they show that increased expression of this gene may be involved in idiopathic pulmonary fibrosis and discuss some mechanistic pathways. However, as they note, this constitutes only the "beginning of an elucidation" and can be construed as weak from a clinical interest point of view, especially since it is the only signal discussed in detail (see H.).

Following careful consideration, we have agreed to remove the HMCN1 example from the resubmission as we agree this story needs more experimental work to reach the levels for publication in this journal. A concern also shared by R4.

F. Suggested improvements: experiments, data for possible revision

1. Improve the presentation and dissemination of their results, ideally through the implementation of a searchable portal (solutions exist for single-point signals, e.g. pheweb). It is especially important that variants constituent of the collapsing tests are documented.

We have released a public portal to accompany our PheWAS data and cite it's link in the revised manuscript (<http://azphewas.com/>). It currently corresponds to gene-level associations for the 200K public exomes, but these data will be updated with the new associations statistics alongside the UK Biobank release schedule.

2. Build upon the discussion of signals, for example through the discussion of positive controls, as well as further signals with translational potential.

We further describe positive control findings from our analyses throughout the revised manuscript: see pg 7, lines 193-198, pg 10 lines 23-235, pg 13 lines 302-305, supp table 9, and lines 304-306).

G. References: appropriate credit to previous work?

0. The authors could refer more to previous rare variant analyses, in particular with respect to the following points:

- which methods did previous papers use to aggregate the effect of rare variants (WES, WGS, and even imputed GWAS)?
- to what extent did the choice of methods place hypotheses on the type of signal that could be detected, and which type of signal was indeed detected?

This can then spur an interesting discussion on how the current dataset can be used to validate these hypotheses, and the methods required for that.

We have added more discussion of rare variant association analysis methods and other large scale studies that have employed them. See pg 17 lines 431 - 438

1. The authors should report, in particular for collapsing tests, whether any of their gene/phenotype association has been previously determined, and if yes in which context (GWAS, previous rare variant, family studies...) and present supporting references.

We provide additional fields in Supplementary Table 8 and Supplementary Table 9 for the collapsing analyses results highlighting the enrichment of positive control relationships amongst our collapsing analyses outputs on the basis of those gene-phenotype relationships already documented in the Online Mendelian Inheritance in Man (OMIM) database.

H. Clarity and context: lucidity of abstract/summary, appropriateness of abstract, introduction and conclusions

0. Presenting data at this scale is challenging, & this article is overall well-written and constructed. However, perhaps as a consequence of the vast scope of analyses performed, it is not entirely clear what the scientific focus of the paper is. Currently it feels neither like a resource paper (data accessibility/browsing/sharing is too limited), nor an in-depth analysis of what Biobank-scale WGS can tell us about the architecture of human disease (discussions on effects of rare variants are too succinct), nor like an examination of the clinical relevance of such analyses (see below).

We anticipate this concern should be addressed with the release of the public facing PheWAS portal to accompany the association statistics generated by this large-scale PheWAS.

1. As mentioned in E., the paragraph about HMCN1 feels somewhat ancillary. This is the 400th signal in terms of strength of association. How did the authors choose it for discussion? Some of the top-associated collapsing tests provide good proof of concepts: ALPL, APOC3, CST3, APOB, GPT... If the authors wish to focus the paper on clinically relevant gene/phenotype associations, the paper would benefit from discussion of those, plus any other additional signals of potential clinical/drug development importance.

We are convinced by R3, R4 and the editors that a proper evaluation into the biology of any of the novel associations requires considerable experimental and translational validation that is outside the scope of this paper. Therefore, we have removed the HMCN1 example to focus on other aspects of the work, while we continue to work on HMCN1 biology over the coming years.

Referee #4:

This manuscript describes a massive analysis linking rare coding variants identified from whole exon sequencing on DNA from 269,171 participants in the UK Biobank to a large series of binary and quantitative traits identified based on clinical and laboratory data. The authors identified a large number of protein truncating variants (PTVs) and missense mutations and analyzed

associations with both individual sequence variants and groups of presumed loss of function variants in the same gene. A large number of known associations were found and there were many new associations identified. The authors make a strong case that the approaches described will could eventually lead to new insights into the molecular bases of human disease and suggest promising drug targets.

One weakness of the current manuscript is the authors effort to highlight a specific example of the utility of their approach. They chose to focus on the new association they identified between a PTV in the gene encoding hemicentin 1 (HMCN1), a secreted component of the extracellular matrix that has been suggested to contribute to the organization of cell adhesions, and an increase in the ratio of FEV1 to FVC. the authors correctly point out that the FEV1/FVC ratio is often increased in patients with restrictive lung diseases, including idiopathic pulmonary fibrosis (IPF). They include convincing data showing that expression of HMCN1 is increased in whole lung RNA from patients with IPF and suggest that this finding validates their approach. However, this change is in the opposite direction of the change suggested by their genetic data. The PTV in HMCN1 would presumable lead to a reduction in functional HMCT1, not the increase that is seen in IPF. This rare variant (with presumed reduction in HMCN1 function) is associated with the increase in FEV1/FVC that the authors used to identify a link to IPF. Of course, there are many other potential explanations for increases in the FEV1/FVC ratio other than restrictive lung disease and the fact that HMCN1 expression is actually increased rather than decreased in IPF suggests that the association identified probably had nothing to do with IPF.

It is therefore unclear why the authors chose to evaluate how HMCN1 expression levels might alter the biology that underlies pulmonary fibrosis. The experiments they include also somewhat miss the point of the effects seen on HNCN1 gene expression seen in IPF. Rather than overexpress HMCN1 (the abnormality seen in IPF lungs) the authors show that knockdown of HMCN1 leads to a reduction in TGFb-induced upregulation of alpha smooth muscle in cultured primary lung fibroblasts. They also show that knockdown causes a reduction in baseline collagen 1a1 expression and claim that this knockdown also inhibits the increase in col1a1 induced by TGFbeta. However, when one takes into account the effects on baseline collagen expression, TGFb seems to induce similar proportionate increases in both control and HMCN1 KD fibroblasts. Again, since if anything, the effects seen are in the opposite direction of what would be required to connect a decrease in functional HMCN1 to an increase in FEV1/FVC, none of these data appear to shed light on how a PVT in the HMCN1 gene would increase the FEV1/FVC ratio.

In summary, this is a very impressive paper that includes identification of many new genetic variants that might contribute to human disease and/or drug response. Unfortunately, the single example the authors chose to evaluate in more detail does not actually strengthen the manuscript and should either be deleted or replaced by a more convincing example, where the direction of effect of the variant and the underlying biology are more congruent.

Taking into consideration the thoughtful feedback from our expert reviewer alongside similar comments from R3 we have made the difficult decision to exclude the HMCN1 example from this manuscript and work on this story independently and with more experimental work over time to

better understand what relevance HMCN1 inhibition has to lung health beyond the physiological phenotypes currently identified.

Reviewer Reports on the First Revision:

Referee #1 (Remarks to the Author):

The authors have done a good job in the revision, well done, but it is not quite there yet.

In response to my previous review, the following issues remain:

1. Choice of statistical test. I like the new para beginning line 211, which has partly dealt with my criticism. However this is going to be a really important point for future studies, and I would encourage the authors to do the SPA vs Exact test across all autosomes not just chr1 (i.e. test the full dataset). Yes it will burn some fossil fuel, but important.

1b. abstract states "The latter revealed 1,703 statistically significant gene-phenotype relationships,"

- please add after the 1703 "versus xxx expected by chance" e.g. by using simulations or some other method. This then helps show the value of what has been found, and contributes to the choice of test discussion.

2 Replication. The authors have partially addressed my previous major comment in the para beginning line 198 and in Supp Table 2 with replication in Finngen. However quite how many signals replicate or not is unclear from the text and Table. I would like at least a sentence or two on this in the main text. Does NA in the Table mean the phenotype was not tested in Finngen, or that the P value was 0.1, or what? More work needed on this please.

Minor:

3. In Fig 3c, in response to my previous comments about 'unbelievable associations' is it possible to label the previously known disease associations in a different colour or something to make them stand out as 'known truth'.

I note some of the other reviewers made the same criticisms as I did about some points. However I have not been over the authors responses to the other reviewers in this round.

regards & signed, david van heel

Referee #2 (Remarks to the Author):

My comments to the authors were addressed satisfactorily in the review. I defer to the remaining reviewers on whether the Fisher's test benchmarking is technically satisfactory.

My only remaining minor comment is to include the processing times in the Methods section of the manuscript, instead of only in the responses.

Referee #3 (Remarks to the Author):

In this revision, the authors made efforts to address my and other reviewers' comments. While the manuscript has been improved, some of my initial concerns remain, particularly regarding the robustness of the statistical test used. Below I reproduce my original comments verbatim, with

additional comments and responses for the authors. For concision, the authors' replies are not included, but my comments refer to them. This is followed by a second section of additional comments.

1. The association methodology, which uses Fisher's exact test for binary traits and a linear regression for quantitative traits, is quite simplistic (see D. below). The use of these unadjusted models should be justified extensively, ideally replaced, or complemented using sensitivity analyses.

The authors misunderstood my comment. I did not suggest rare variants should be ignored in a WES study (!). I remarked that the method used for both single-point and collapsing tests (Fisher's) is very rudimentary considering the Biobank-oriented method development efforts of late. For comments on the authors' comparison with SAIGE, see below.

3. Section 6. of the methods describes sample filtering based on ethnicity, excluding all non-European samples. While it is understood that this was done to alleviate the risk of spurious associations, (a) these samples could have been included had the authors chosen to adjust for ethnicity using a more complex model, (b) the authors themselves show these samples exhibit novel, specific and disease-relevant variants in Fig. 1g, and (c) excluding non-Europeans in the first paper to use the UKB WES data is not ideal. However, I do acknowledge that the cumulative proportion of non-Europeans does not exceed ~10%(Fig 1.f).

The authors should add their reply: "Methods such as SAIGE and BOLTLMM, although they allow for inclusion of PCs, have not been used or tested for transethnic analysis. The appropriate approach for such analysis would be to analyse each ethnicity separately and then combine the results in a meta-analysis framework. In the UK Biobank instance, we expect such results to be overwhelmed by the European representation, and therefore, the non-European ancestry groups will add little information in this context. We feel this is best addressed by the generation of equivalent medical research resources in non-European collections" to the main text. Although the documentation of SAIGE states that multi-ethnic associations are untested, they also suggest that it may be well-powered and controlled with the adjustments of PCs. Without these precisions, their added paragraph "Furthermore, the recognised Eurocentric bias across the field of human genomics has ethical and scientific consequences, exacerbating genomics healthcare inequality and limiting power to identify novel associations. The need to establish linked genomic and phenotypic resources equivalent to the exemplar UKB standard in other global populations has never been clearer" reads somewhat hypocritical, as restricting the analysis to unrelated white British individuals in a diverse cohort (mainly because of the chosen method, which is in turn to mitigate compute costs, i.e. convenience), can be perceived as perpetuating the Eurocentric focus of human genetics, rather than exploiting the little non-European information that does exist.

4. The Data Availability statement does not state whether full exWAS and variant collapsing summary statistics will be publicly available. They should. Given the scope of the paper (phenome-wide single-point exWAS and rare variant collapsing tests), and given the fact that it is the first such analysis of the UKB Exomes dataset, results should be disseminated widely, and results should be query-able by researchers, e.g through a searchable online portal. Variants contributing to every associated collapsing signal should be listed along with their single-variant effects. The presentation of association results, currently in the form of Table 1 (which feels more like a supplementary table) and ST6, could be improved (e.g., reporting only the strongest p-value for gene/phenotype pairs, improving phenotype descriptions, etc).

I would like to commend the authors on implementing a browsable portal. One small comment is that the collapsing models are not documented in the results table when browsing. It would be good to display the details in an info bubble.

0. The authors mention a Fisher's as the main association test for single-point analyses. This

approach is historical and does not correct for well-documented sources of inflation/T1E. I understand the authors wished to include very low allele counts, however the overall impact of using an uncorrected method outweighs the single advantage of being able to include very rare variants in single-point testing. The reason variant aggregation tests exist is to bypass the fundamental limitations of single-point methods at very low allele counts. If it is retained (I don't think it should), Fisher's testing should only be used for those very rare variants. For anything not ultra-rare, up-to-date methods adjusting for ethnic and other covariates, as well as random relatedness effects, should be used. The size of the study is not a justification either, as single-point GWAS studies of the full imputed UK Biobank, which are comparable in terms of sample size and variant numbers, have successfully used subtler models. Using a better model would make it unnecessary to remove non-Europeans and unrelated individuals.

I thank the authors for augmenting their article with the SAIGE comparison. I must say that the correlation they show is impressive. This would suggest that in the current setting, a more complex regression model is "not needed" versus Fisher's. Given that this is a conclusion with significant impact on the community, I feel it has to be fully irrefutable. To that effect, I have several comments below.

- The authors perform additional preparation steps to their data to make it match the assumptions of Fisher's (e.g. rebalancing sex-specific c/C ratios since they cannot adjust for sex). These operations are computationally costly, and should be added to the cost of running their analysis vs. SAIGE to avoid an unfair comparison.
- SAIGE is notoriously inefficient versus REGENIE when it comes to multi-phenotype associations. Please repeat the comparison for REGENIE and include running time.
- REGENIE 2 also performs variant aggregation tests natively. SAIGE is also able to do it if fed aggregated genotypes. Please include this in your comparison.
- In all supplementary figures relating to this comparison, the authors select predictors that pass a given threshold in both studies (according to the legends). What if a test produces $p=0.05$ with one method and $p=1e-100$ in the other? These will reduce correlation and should be included.
- What motivated the choice of the phenotypes? Wouldn't one expect increased correlation among "all chapter IX" variables? What is the effective number of traits actually being compared?
- Another enlightening metric could be a comparison of lambdas between the different methods.
- Please also report r in general, not in subsets of variants
- In the supplementary figures, give the coefficients of the blue regression lines
- The authors repeatedly say that "Fisher's exact is more robust for lower frequencies". Does this simply mean that the p -values are less strong? In the absence of a ground truth, is this really an argument for Fisher's superiority? Fisher's exact is recommended in place of chi-squared in the analysis of 2x2 tables when individual cell counts are smaller than 5 or 10. To expand this very specific rule of thumb to "Fisher's is better than a regression model when analysing rare variants" needs much more justification than currently provided in this article.

1. A linked point is that some of the lambdas (inflation factors) are very low or very high. This suggests that a non-negligible part of the test statistics and resulting P-value distributions may be poorly calibrated. I would suggest using hard lambda thresholds (e.g. 0.95 - 1.05) to exclude these analyses from further consideration.

Thanks for clarifying that most of the abnormal lambdas come from the recessive model. Please report lambdas in the 0.95-1.05 range and separately for the different models.

2. Rare variant testing. Again, it is surprising to read that the authors have used a Fisher's exact test for rare variant association. As above, the same questions about adjustment of spurious effect apply here. Arguably, the most interesting use of WES is the examination of rare variants. The authors use a collapsing test, which makes important hypotheses regarding the architecture of the underlying signals. The authors mention in passing that they frequently observe concordant directions of effect in genes with multiple associated variants, but this and its relationship to the collapsing test should be discussed rigorously. It is regrettable that the authors did not examine

different architectures, such as those modelled by SKAT-type methods, or SKAT-O optimal tests. In particular, I would have liked to see a distribution of the rho parameter, which could have confirmed the authors' hypotheses regarding signal architecture. These methods have historically scaled poorly, however implementations now exist for biobank sized datasets and the authors should use them.

Thank you for adding this discussion. Since it is relevant to the choices made in the paper, I would include the paragraph starting at "The assumptions of the burden test..." in your reply to the text.

3. Significance threshold. The authors are convincing in their calculations, however I would like to see their threshold confirmed by a calculation that would take into account the effective number of traits, variants and analyses through reduction of the respective correlation matrices (phenotype, LD and z-scores). It would be good if the authors added more details about their permutation method. Finally, in this paper the authors report results from two correlated analyses using the same test, single-point and collapsed. Two different thresholds were used instead of a single study-wide reporting threshold. The authors should convincingly justify this, or use a single threshold that adjusts for the increased reporting burden caused by the collapsing analysis. The added details around the permutation are very helpful, thanks to the authors for adding them.

In their reply, the authors say "the ExWAS and collapsing represent considerably different test statistic distributions and we [consider them separately]". I disagree. The two tests are the same (Fisher's), test two different, but very closely related hypotheses, to the extent that one can be considered an extended scenario of the other. The tests are also not independent. As far as I understand, the exWAS contains test statistics for all variants with $MAC > 5$. Imagine a case with 2 qualifying RVs in the collapsing part. One of them with $MAC = 5$ and the other with $MAC = 100$. The collapsed test will be very similar to the single-point test of the second variant.

I maintain that a single threshold would truly be "study-wide" here. Authors could use the aggregation p-value ($2e-9$) to declare significance across the board.

ADDITIONAL COMMENTS.

Line 80: Is it really well-recognised, or do these variants "need" to have large effects to be detected? It is an issue of detectability, not necessarily biological truth.

Line 176: Mention here, early on, the testing methodology used.

Line 275: MTR has not been defined or explained before.

Line 341: I am confused by this paragraph. Does the fact that the authors find genes associated with haematological malignancies necessarily mean that they are picking up somatic variations vs. a germline susceptibility?

Line 356: Mention here the author's own comparison of fold increase vs. the 50k using their own methodology. Is there a marked difference, and can the authors expand on methodological differences between the cited studies and theirs?

Line 361: This is a "dangerous" paragraph. It is of course statistically expected that there will be false negatives using a stringent P threshold. It does not mean that people should start investigating "suggestive" signals like in the early (and bad) days of GWAS. I would recommend moving this to the supplementary.

Line 369: Does this mean that 18% of genes associated with a phenotype in the collapsing analysis also contain a significant variant in the single-point analysis? An important question is whether one signal is "tagging" the other in these cases, which could be assessed through conditional testing – were covariates allowed in the testing model.

Lines 381-386: This newly added paragraph is quite confusing.

Line 386-393: This is quite interesting. Was this accompanied by loss of power for some signals (i.e. do the reverse) ? Do the authors then recommend including more common variants in collapsing analyses?

Line 443-445: This is not a sentence.

Line 442: This newly added paragraph feels out of place in the wider flow of the article. Would it feel more at home in the intro?

Author Rebuttals to First Revision:

Dear Editor and Reviewers:

We would like to thank the editorial team and reviewers for their comments and the opportunity to further improve our manuscript. We have now run SAIGE SPA for variants across all autosomes for all Chapter IX phenotypes. Consistent with our previous analysis limited to chromosome 1, we found that study-wide significant signals were highly correlated between SAIGE and the Fisher's exact test ($r = 0.99$) and $r = 0.95$ across the full p -value distribution.

We also appreciate the request to incorporate analyses of non-European ancestries. We are in complete ethical agreement that we should incorporate these individuals and set the right example for the community, irrespective of power expectations. We thus ran an additional PheWAS for each major non-European ancestral group with at least 1,000 exome sequenced participants (South Asian, East Asian, African). As expected, no novel gene-phenotype reached significance in any single population. However, we also performed a "pan-ancestry" analysis in which we used a Cochran–Mantel–Haenszel (CMH) test to generate a combined, stratified p -value representing the data from all four genetic ancestry strata, including the European cohort. We also ran a pan-ancestry PheWAS for quantitative traits. These results are summarized in the manuscript and in Figure 4.

Below, we address each of the specific editorial and reviewer comments and cite the location of the edits corresponding to the marked-up version of our resubmission.

Referee #1

1. Choice of statistical test. I like the new para beginning line 211, which has partly dealt with my criticism. However this is going to be a really important point for future studies, and I would encourage the authors to do the SPA vs Exact test across all autosomes not just chr1 (i.e. test the full dataset). Yes it will burn some fossil fuel, but important.

In the revised text we have expanded the Chapter IX comparison across all autosomes. Our conclusions are the same as those based on chromosome 1 alone, in that results are highly correlated ($r=0.99$) with Fisher's exact p -values being more conservative as the allele frequency reduces. We have updated the relevant text, figures and tables accordingly throughout (Supplementary Methods: Comparing exWAS results using Fisher's exact test versus SAIGE with covariates and main manuscript Pg 9, lines 229 – 234).

1b. abstract states "The latter revealed 1,703 statistically significant gene-phenotype relationships," - please add after the 1703 "versus xxx expected by chance" e.g. by using simulations or some other method. This then helps show the value of what has been found, and contributes to the choice of test discussion.

We have been conservative in setting our study-wide significance threshold using both the empirical synonymous and the permutation-based null distributions. As a result, at that conservative threshold the expected number is zero to nearest integer. We have added some text in the methods to emphasise the interpretation of the selected adjusted p-value threshold (See pg 12, line 279).

2 Replication. The authors have partially addressed my previous major comment in the paragraph beginning line 198 and in Supp Table 2 with replication in FinnGen. However quite how many signals replicate or not is unclear from the text and Table. I would like at least a sentence or two on this in the main text. Does NA in the Table mean the phenotype was not tested in FinnGen, or that the P value was 0.1, or what? More work needed on this please.

Indeed, FinnGen studied approximately ~2.8K phenotypes compared to over 18,000 in the UK Biobank. Thus, some phenotypes in the UKB don't have a comparison in FinnGen. Because of the different ontologies adopted by FinnGen and UKB, it is not a simple exercise to map all phenotypes across the resources to differentiate whether (i) a relevant phenotype wasn't studied in FinnGen or (ii) a relevant phenotype was studied in FinnGen but the p-value of the association is $p > 10^{-4}$. We have, however, taken this opportunity to update the FinnGen exercise using the most recent public release FinnGen r5 and updated the text, tables and figures accordingly. For each variant in Table S2 (missense and PTV tabs), we have checked if any variant in that gene is associated with a related phenotype at $p < 10^{-4}$. If not, then we label it as 'NA'.

Minor:

3. In Fig 3c, in response to my previous comments about 'unbelievable associations' is it possible to label the previously known disease associations in a different colour or something to make them stand out as 'known truth'.

We appreciate this suggestion. The majority of the study-wide significant associations are well-established and agree it is important to communicate this information. As such, we have revised supplementary tables 8 and 9 to provide greater clarity with respect to associations that are well established in OMIM. We believe that the incorporation of this information in supplementary tables 8 and 9 provides the reader with a more comprehensive overview of prior knowledge, and facilitates greater understanding beyond what could be communicated through the annotation of Fig 3c.

Referee #2:

My comments to the authors were addressed satisfactorily in the review. I defer to the remaining reviewers on whether the Fisher's test benchmarking is technically satisfactory. My only remaining minor comment is to include the processing times in the Methods section of the manuscript, instead of only in the responses.

We now include computational requirements in the Methods section of revised manuscript (See Pg 28, lines 729 – 743).

Referee #3:

The authors should add their reply: “Methods such as SAIGE and BOLTMM, although they allow for inclusion of PCs, have not been used or tested for transethnic analysis. The appropriate approach for such analysis would be to analyse each ethnicity separately and then combine the results in a meta-analysis framework. In the UK Biobank instance, we expect such results to be overwhelmed by the European representation, and therefore, the non-European ancestry groups will add little information in this context. We feel this is best addressed by the generation of equivalent medical research resources in non-European collections” to the main text. Although the documentation of SAIGE states that multi-ethnic associations are untested, they also suggest that it may be well-powered and controlled with the adjustments of PCs. Without these precisions, their added paragraph “Furthermore, the recognised Eurocentric bias across the field of human genomics has ethical and scientific consequences, exacerbating genomics healthcare inequality and limiting power to identify novel associations. The need to establish linked genomic and phenotypic resources equivalent to the exemplar UKB standard in other global populations has never been clearer” reads somewhat hypocritical, as restricting the analysis to unrelated white British individuals in a diverse cohort (mainly because of the chosen method, which is in turn to mitigate compute costs, i.e. convenience), can be perceived as perpetuating the Eurocentric focus of human genetics, rather than exploiting the little non-European information that does exist.

We are in complete agreement that we should incorporate the non-European ancestry individuals and set the right example for the community, irrespective of statistical expectations. We have now expanded our study by incorporating the signals from 4,744 African, 1,475 East Asian and 5,714 South Asian genetic ancestry participants for the set of 4,836 binary traits with at least 5 cases in at least one of the non-European ancestries. Alongside this we generated pan-ancestry collapsing PheWAS results for all quantitative traits. See Page 17 line 406 – page 19 line 463.

I would like to commend the authors on implementing a browsable portal. One small comment is that the collapsing models are not documented in the results table when browsing. It would be good to display the details in an info bubble.

The collapsing models are documented in the global filters of the portal – we’d like to thank the reviewer for highlighting that this is not clear from the table itself. We will modify the info bubble to inform users where they can find this documentation in our next production release.

I thank the authors for augmenting their article with the SAIGE comparison. I must say that the correlation they show is impressive. This would suggest that in the current setting, a more complex regression model is “not needed” versus Fisher’s. Given that this is a conclusion with significant impact on the community, I feel it has to be fully irrefutable. To that effect, I have several comments below.

- The authors perform additional preparation steps to their data to make it match the assumptions of Fisher's (e.g. rebalancing sex-specific c/C ratios since they cannot adjust for sex). These operations are computationally costly, and should be added to the cost of running their analysis vs. SAIGE to avoid an unfair comparison.

The pre-processing steps referred to by the reviewer are included in our previous estimates around construction of phenotype matrices. We have now added some text to the compute section to clarify this on Pg 28, lines 729 – 743. We'd like to highlight that these are performed once per binary phenotype and then applied to all variants and genes studied for that binary phenotype. For example, the rebalancing case-control sex ratio is performed once per phenotype, resulting in a total of 17,362 tests, which is negligible in comparison to the ~108 billion variant-based ExWAS and ~4.5 billion gene-based collapsing binary tests performed in this study.

- SAIGE is notoriously inefficient versus REGENIE when it comes to multi-phenotype associations. Please repeat the comparison for REGENIE and include running time.

- REGENIE 2 also performs variant aggregation tests natively. SAIGE is also able to do it if fed aggregated genotypes. Please include this in your comparison.

In this paper, we have not introduced a novel untested statistical framework. Rather, we have focused on applying a conventional exact test in a carefully designed (harmonised) experimental setting. Surprisingly, exact test implementations have not been routinely included in benchmarking of novel regression-based methods.

Viewed holistically and conservatively, the results represented in our study do not suggest concern for a high Type I error rate that would necessitate additional methodological evaluation. In response to the reviewer's requests during the first round of revisions, we compared to SAIGE as the most used framework in biobank settings for the past two years. Although we agree that a study comparing the growing number of available statistical approaches would be of interest, we feel that this is tangential to the scope of our current manuscript, particularly in the absence of concerns over QC/robustness. With the UKB exome data now publicly available, we highlight in the discussion (Pg 20, lines 494-501) that future work should focus on benchmarking both novel and conventional algorithms on these data.

For the purposes of the response, in accordance with the author recommendations, we have now run the additive model using the approximate Firth likelihood ratio test with REGENIE (switching to the Firth likelihood ratio test for p -values < 0.01) to compare to the FET results. We restricted this comparison to chromosome 1 for the 324 traits in Chapter IX, as our FET-SAIGE comparison highlighted chromosome 1 is a fair representation of autosome-wide results, and thus the additional compute is not warranted. For the chromosome 1 comparison, we adjusted REGENIE with the same covariates as SAIGE, i.e., age, sex, sequencing batch and 10 PCs as provided by Bycroft et al. Following additional advice from the REGENIE website, we restricted comparisons to only variants with $MAC \geq 5$.

*Across the 324 traits, for signals achieving a $p < 1 \times 10^{-8}$ across any of Fisher's Exact, SAIGE or REGENIE, the Pearson's correlation coefficient of the $-10 * \log_{10} p$ values between REGENIE and FET is 0.999 with*

95% confidence interval 0.998-0.999. Pearson's correlation coefficient of the $-10 \cdot \log_{10} p$ values between SAIGE and FET is 0.999 with 95% confidence interval 0.9987-0.9995. See Figure 1 in this response.

Across the 324 traits on chromosome 1, there are 62,222,510 variant-trait pairs where REGENIE, FET and SAIGE SPA return a p-value (excluding where SAIGE SPA $p_v=0$). Pearson's correlation coefficient on the $-10 \cdot \log_{10} p$ values between REGENIE and FET is 0.938 with 95% confidence interval 0.93799-0.93805. Pearson's correlation coefficient on the $-10 \cdot \log_{10} p$ values between FET and SAIGE is 0.9497 with 95% confidence interval 0.94968-0.94973.

When we compare the p-values for different allele frequency bins, we observe that all three methods provide similar p-values for common and low frequency variants, with both REGENIE and SAIGE generating increasingly lower p-values for rarer variants ($MAF < 0.001$). See Figure 2 in this response.

Regarding run time, for the chromosome 1 Chapter IX collection, REGENIE ran the additive model ($n=62,222,812$ statistical tests) 1.08 times faster compared to the time it took for Fisher's exact to complete all three genetic models in one analysis ($n=205,072,560$ statistical tests). REGENIE does not currently have the option to run 3 models in one analysis.

Figure 1

Figure 2

- In all supplementary figures relating to this comparison, the authors select predictors that pass a given threshold in both studies (according to the legends). What if a test produces $p=0.05$ with one method and $p=1e-100$ in the other? These will reduce correlation and should be included.

Across the autosomes for the 324 traits, there are ~655M variant-trait pairs where both FET and SAIGE return a p -value. Due to the large number of pairs, it is challenging to create plots and therefore for plotting only, we focus on the subset of 2,105,332 variant-trait pairs with p -values < 0.01.

Specifically to the reviewers concern, the correlation when considering all results achieving a $p < 1 \times 10^{-8}$ in **either** Fisher's **or** Saige SPA is 0.99 (now Supplementary Figure 2). There are no extreme p -value discordance examples after exclusion of SAIGE SPA $p=0$ from comparisons. See "Supplementary Methods: Comparing ExWAS results using Fisher's exact test versus SAIGE with covariates" and main manuscript pg 9, lines 229 – 234.

- What motivated the choice of the phenotypes? Wouldn't one expect increased correlation among "all chapter IX" variables? What is the effective number of traits actually being compared?

This was motivated by our Centre's particular interest in cardiovascular disease. As also requested by R1, to get more independent data points in this benchmarking we have now expanded the Chapter IX assessment to all autosomes rather than just chromosome 1.

In response, we have now calculated pairwise correlations for all 324 Union phenotypes included in Chapter IX. Among 52,326 phenotype pairs, 71 pairs achieved an $r^2 > 0.4$ and 140 pairs achieved an $r^2 > 0.2$, suggesting a subset—but not the majority—of these phenotypes are strongly correlated. For example, setting a phenotype pairwise correlation pruning threshold of $r^2 < 0.4$ results in 262 (80.1%) effective phenotypes of the 324 Chapter IX Union phenotypes adopted for these comparisons. We include these correlation statistics in the revised supplemental methods describing this activity and have included the list of pairs with an $r^2 > 0.2$ as supplementary table 29.

- Another enlightening metric could be a comparison of lambdas between the different methods.

We have provided this in the revised manuscript for the expanded comparison of all autosomes in Chapter IX. For Fisher's we observed a median lambda of 1.00064 [range 0.967544-1.069814] and for the SAIGE SPA we observed a median lambda of 0.995256 with a wider lambda distribution range [range 0.937221 – 1.093990]. Consistent with the high correlations described earlier between the two Phred distributions (See pg 9, lines 233 – 236).

- Please also report r in general, not in subsets of variants

Across the autosomes for the 324 traits, there are 654,927,125 variant-trait pairs where both FET and SAIGE return a p-value. Pearson's correlation coefficient across all variant-trait pairs was $r = 0.95$ and we have added this statistic to the manuscript (See pg 9, line 229). Due to the large number of pairs, it is difficult to create plots and therefore for plotting only, we focus on the subset of 2,105,332 variant-trait pairs with p-values < 0.01 . (See "Supplementary Methods: Comparing exWAS results using Fisher's exact test versus SAIGE with covariates").

- In the supplementary figures, give the coefficients of the blue regression lines

This is now provided in the revised supp figures.

- The authors repeatedly say that "Fisher's exact is more robust for lower frequencies". Does this simply mean that the p-values are less strong? In the absence of a ground truth, is this really an argument for Fisher's superiority? Fisher's exact is recommended in place of chi-squared in the analysis of 2x2 tables when individual cell counts are smaller than 5 or 10. To expand this very

specific rule of thumb to “Fisher’s is better than a regression model when analysing rare variants” needs much more justification than currently provided in this article.

We reviewed the text and identified the two relevant sections. The language in the manuscript is consistent with the interpretation of the comparisons and the reviewer’s own view. For convenience, we provide below the language in the two relevant sections (one in main and one in supplemental). Note, we could not find use of “is better than.”

Main document (pg 9, lines 231 – 233). “The Fisher’s exact p-values were more conservative than SAIGE for lower frequency variants (MAF ≤ 1%) (Supplementary Figures 3 and 4, Supplementary Table 6).”

*Supplemental document: “When focusing across different MAF ranges: common (MAF≥0.05), low frequency (0.01≤MAF<0.05), rare (0.001≤MAF<0.01) and very rare (MAF<0.001) variants (Supplementary Figure 4a-4d), SAIGE reports lower p-values than the Fisher’s exact test p-values with decreasing MAF. **We therefore reach the conclusion that in this setting the Fisher’s exact p-values are more conservative than SAIGE.**”*

In both situations we believe that “more conservative than” captures the reviewer’s view that Fisher’s generates less impressive p-values in the low frequency range.

Thanks for clarifying that most of the abnormal lambdas come from the recessive model. Please report lambdas in the 0.95-1.05 range and separately for the different models.

*The full set of lambdas for every studied phenotype-model combination are available in Supplementary Table 22 – lambda distributions. We have created a new supplemental table below that includes the percentage of lambdas falling between lambda ranges, per model introduced as new **Supplementary Table 23.***

Model	Binary traits		Quantitative traits	
	λ 0.95-1.05 (%)	λ 0.90-1.10 (%)	λ 0.95-1.05 (%)	λ 0.90-1.10 (%)
flexdmg	16097 (93)	17338 (100)	1278 (90)	1342 (95)
flexnonsyn	16387 (94)	17336 (100)	1301 (92)	1357 (96)
flexnonsynmtr	16200 (93)	17337 (100)	1298 (92)	1367 (97)
ptv	14025 (81)	17057 (98)	1248 (88)	1338 (95)
ptv5pcnt	14343 (83)	17131 (99)	1238 (87)	1327 (94)
ptvaredmg	15895 (92)	17338 (100)	1283 (91)	1354 (96)
raredmg	15525 (89)	17316 (100)	1311 (93)	1388 (98)
raredmgmtr	14912 (86)	17218 (99)	1321 (93)	1402 (99)
rec	11866 (68)	16086 (93)	1216 (86)	1407 (99)
syn	15868 (91)	17331 (100)	1389 (98)	1416 (100)
UR	14548 (84)	17174 (99)	1273 (90)	1381 (98)
URmtr	14177 (82)	17013 (98)	1272 (90)	1386 (98)

In their reply, the authors say “the ExWAS and collapsing represent considerably different test statistic distributions and we [consider them separately]”. I disagree. The two tests are the same (Fisher’s), test two different, but very closely related hypotheses, to the extent that one can be considered an extended scenario of the other. The tests are also not independent. As far as I understand, the exWAS contains test statistics for all variants with $MAC > 5$. Imagine a case with 2 qualifying RVs in the collapsing part. One of them with $MAC = 5$ and the other with $MAC = 100$. The collapsed test will be very similar to the single-point test of the second variant. I maintain that a single threshold would truly be “study-wide” here. Authors could use the aggregation p-value ($2e-9$) to declare significance across the board.

We have revised the threshold for exWAS to $p \leq 2 \times 10^{-9}$ throughout the manuscript and updated all tables and figures accordingly. (See Pg 26, lines 688 – 692).

ADDITIONAL COMMENTS.

Line 80: Is it really well-recognised, or do these variants “need” to have large effects to be detected? It is an issue of detectability, not necessarily biological truth.

We have revised the text accordingly (See pg 3, lines 81 – 82).

Line 176: Mention here, early on, the testing methodology used.

We have revised the text accordingly (See pg 8, lines 188-193).

Line 275: MTR has not been defined or explained before.

We have revised the text accordingly (See pg 12, lines 286-288).

Line 341: I am confused by this paragraph. Does the fact that the authors find genes associated with haematological malignancies necessarily mean that they are picking up somatic variations vs. a germline susceptibility?

*Aside from DDX41, which is known familial autosomal dominant form of myeloproliferative neoplasms, the remaining gene signals are enriched for established somatic-driven haematological malignancies. However, we agree the importance to make that data more accessible to our readers. To this effect, we have included a new **Supplementary Figure 6** that plots the distribution of the alternative allele ratio (i.e., the percentage of all reads in the sequence data at the site that support the alternative allele) and include as a reference the established germline BRCA1 and BRCA2 alternative allele ratios, which are known germline disease drivers. We now cite this supplemental figure in the main manuscript (See pg 15, line 358).*

Line 356: Mention here the author’s own comparison of fold increase vs. the 50k using their own methodology. Is there a marked difference, and can the authors expand on methodological differences between the cited studies and theirs?

We have introduced the fold increase as function of sample size growth from 50K to 300Kv1 (i.e., prior to the 300Kv2 phenotypic refresh) (See pg 15 line 373 – pg 16 line 377). When comparing the results of our collapsing method applied to the full UKB dataset versus the first tranche of 50K data, we observed an 18-fold increase in statistically significant gene-trait associations. Incorporating the updated phenotypic data for the same set of samples released up to July 2020 resulted in a 24-fold increase in significant associations compared to the 50K data.

An expanded description of the methodological differences between the 50K comparison studies can be found in “Supplementary Methods: Comparing 50K UKB gene-level results across multiple studies”.

Line 361: This is a “dangerous” paragraph. It is of course statistically expected that there will be false negatives using a stringent P threshold. It does not mean that people should start investigating “suggestive” signals like in the early (and bad) days of GWAS. I would recommend moving this to the supplementary.

We have moved this section to the supplement.

Line 369: Does this mean that 18% of genes associated with a phenotype in the collapsing analysis also contain a significant variant in the single-point analysis? An important question is whether one signal is “tagging” the other in these cases, which could be assessed through conditional testing – were covariates allowed in the testing model.

*This highlights that for 18% of gene aggregate (collapsing) signals an individual variant in those genes (exWAS) also achieved significance. One possible explanation is that this is driven by genes where both common and rare risk variants contribute to disease burden. We now include a citation to our other study currently available in biorxiv (<https://doi.org/10.1101/2020.12.10.419663>) where we performed a deep dive on FLG and asthma risk – finding that two well-known common PTVs and the collection of rare PTVs contribute to increased early-onset asthma risk, and achieve comparable effect sizes (common PTV #1 [rs61816761-G-A] $p = 6.9 \times 10^{-21}$, **OR=1.9**, 95%CI=1.7-2.2; common PTV #2 [rs558269137-CACTG-C] $p = 4.0 \times 10^{-20}$, **OR=1.9**, 95%CI=1.7-2.1; aggregate collection of rare FLG PTV variants not inclusive of two aforementioned common PTVs [$p=2.7 \times 10^{-10}$, **OR=1.7**, 95%CI=1.5-2.1]).* See pg 19 lines 475-479.

Lines 381-386: This newly added paragraph is quite confusing.

We have revised the paragraph to achieve more clarity.

Line 386-393: This is quite interesting. Was this accompanied by loss of power for some signals (i.e. do the reverse) ? Do the authors then recommend including more common variants in collapsing analyses?

It is very feasible to envision running additional collapsing models with more liberal MAFs beyond the PTV class. We have included additional text reflecting some soft recommendation that, in our experience, the collapsing approach is best suited for aggregation of rare variants, and that for interests in more common variants, one might be better placed to rely on the single-point analyses. We did make an exception for the PTV class, both out of curiosity for this exercise, but also because as a variant class we had higher confidence, in general, of their predicted functional effect than what we do for missense variants. See pg 16, lines 401 – 404

Line 443-445: This is not a sentence.

We have revised the text accordingly.

Line 442: This newly added paragraph feels out of place in the wider flow of the article. Would it feel more at home in the intro?

We have moved the text accordingly.

Reviewer Reports on the Second Revision:

Referee #1 (Remarks to the Author):

I have no further comments to be addressed. I would be happy for this study to be published in

Nature should the Editors so decide.

I have not cross-checked that other reviewers comments have been addressed.

The authors are to be commended in this revision for also updating to Finngen v5 results. This new sentence in discussion is also appropriate "Importantly, use of the exact test requires extremely careful quality control, case-control harmonization, and ancestry pruning."

david van heel

Referee #3 (Remarks to the Author):

The authors have now made substantial changes to the manuscript to address mine and other reviewers' comments. These include 1) a comparison of their p-value distribution with that obtained from state-of-the-art methods, and 2) the inclusion of non-European ancestry samples. These changes have improved the manuscript, and I am not convinced that the results presented here are reasonably exempt of type-1 error.

However, upon rereading the manuscript as a whole, I still cannot give a positive review for publication in Nature.

The first reason remains the choice of method. Despite showing that their results are robust in this particular case, analysing without a linear mixed model requires careful pre-processing and testing to eliminate all potential sources of bias, as the authors themselves describe. The use of biobank-ready linear mixed model software, which straightforwardly corrects for this type of bias, should be encouraged, especially since the time advantage of using simpler models will likely disappear when using recent versions such as REGENIE 2. I therefore think that giving widespread exposure to a manuscript that, at least partially for reasons of expediency, chooses a naive, hard to use and therefore error-prone model, could set a bad example for the community.

The second reason is the scope of the paper. Although it presents some insight into the overlap of associations with drug targets, the main output of the paper is a database of PheWAS associations across a very wide range of phenotypes. Without a deeper examination of the role of these variants and the translational potential of these signals, this reads like a resource paper comparable to the Global Biobank Engine (McInnes et al., Bioinformatics, 2018, based on de Boever et al., Nat Commun 2018), or the unpublished Neale results. A paper of higher ambition should go further than, as the authors write, "expand the catalogue of statistically significant associations".

The paper would also benefit from more concise writing. In its current state it quite often gets lost in detail, reads rough and can seem unfocused at times. It would also be nice to make sure the results can be reproduced. Is the code available, have the authors packaged it in a way that allows reproduction and reuse, in particular with respect to statistical testing?

I. 86. Although TOPMed is likely to improve imputation accuracy for RVs due to its size, HRC imputation doesn't allow to go too low in the allele frequency spectrum.

I. 94-96. Please review the grammar of this sentence. This is the place to spend a bit more time introducing the various methods of aggregation and the statistical frameworks involved.

I. 193-198. The reason for this choice (computational efficiency) should be clearly stated from the start. The fact it turned out to be conservative doesn't need to be mentioned here.

I. 201-202. This observation indicates that the authors primarily conducted a frequency-agnostic GWAS, then noting that 26% of signals came from variants with $MAF < 0.5\%$. Focusing on rare variants from the get-go would be better. The authors can comment on the overlap between their common-variant signals and previously reported ones.

I. 214-216. Is this for rare significant PTVs/missense only or irrespective of MAF? How many of these gene/phenotype or indeed variant/phenotype relationships have been found by previous imputation-based UKB studies (e.g. Neale)? In general (not just for PTV/missense), a comparison with imputed GWAS in UKB is important both for novelty analysis and sensitivity. Perhaps a summary here with a couple of sentences and a more extended paragraph in the discussion?

paragraph I. 227. This should be moved to the discussion. Remove runtime comparisons from the main text as the time advantage is expected and comes at the expense of robustness.

I. 288-289. Remove "appropriate". Please reformulate "At this very..." by e.g. "Under this threshold, no positive associations are expected under the null."

I. 307. "in this study"? Do you support/recommend the use of MTR in general as a further filter to select variants more likely to be functional?

I. 370-372. I am still not convinced by this somatic variant paragraph. The added sentence does not particularly help. I suggest to substantially expand on these claims or take it out.

I. 435-442. Remove the suggestive association.

I. 441, 471. "insignificant" is not the proper wording. In Fig 4 legend "landed above" is also not serious wording.

Fig. 4. b. and d. seem quite surprising to me. I would expect a marked asymmetry between blue signals (not significant before inclusion of non-Europeans) and orange (used to be significant in Europeans only), with more of the former, and weaker delta-phred values for the latter. It seems very surprising that adding/removing ~5% of samples changes some p-values by a log-factor of 40?

I 491-496. The proper way to test for this is through a conditional analysis.

I 501-518. Emphasise that this was chosen to improve computational efficiency but is not something people should do in general.

I. 551. What method did the authors "introduce"?

I. 681. All software should be referenced. Rephrase I. 684-686 for clarity.

I. 701. Remove "to account for large case-control imbalances". Also remove "permit the study of extremely rare variants". If that were the reason the authors could have used this test only for those variants with $MAC < 7$.

I.710-712. Doesn't this contradict the "zero expected positives under the null"? In general the number of tests performed seems to change throughout the manuscript, but if the authors performed 108bn variant+4.5bn collapsing, the total exceeds the 38.7 tested, bringing the expected number (under permutation) to over 36 false positives?

I. 788. is this a second paragraph on n of 1 permutation?

I. 794. Combining it with I. 710, does this mean that 18 permuted p-values were between $1.9e-9$ and $2e-9$?

Author Rebuttals to Second Revision:

Below, we address each of the specific editorial and reviewer comments and cite the location of the edits corresponding to the marked-up version of our resubmission.

Referee #1

I have no further comments to be addressed. I would be happy for this study to be published in Nature should the Editors so decide. I have not cross-checked that other reviewers comments have been addressed. The authors are to be commended in this revision for also updating to Finngen v5 results. This new sentence in discussion is also appropriate "Importantly, use of the exact test requires extremely careful quality control, case-control harmonization, and ancestry pruning."

We have ensured this critical sentence persists in the discussion.

Referee #3

The authors have now made substantial changes to the manuscript to address mine and other reviewers' comments. These include 1) a comparison of their p-value distribution with that obtained from state-of-the-art methods, and 2) the inclusion of non-European ancestry samples. These changes have improved the manuscript, and I am now convinced that the results presented here are reasonably exempt of type-1 error. However, upon rereading the manuscript as a whole, I still cannot give a positive review for publication in Nature.

The first reason remains the choice of method. Despite showing that their results are robust in this particular case, analysing without a linear mixed model requires careful pre-processing and testing to eliminate all potential sources of bias, as the authors themselves describe. The use of biobank-ready linear mixed model software, which straightforwardly corrects for this type of bias, should be encouraged, especially since the time advantage of using simpler models will likely disappear when using recent versions such as REGENIE 2. I therefore think that giving widespread exposure to a manuscript that, at least partially for reasons of expediency, chooses a naive, hard to use and therefore error-prone model, could set a bad example for the community.

As previously described by the reviewer, the Fisher's exact test is simpler, significantly more accessible to implement than the comparative software suites, and can provide robust results. The pre-association harmonisation process adopted here, does not substantially deviate from established best practice QC that would be typically implemented when adopting linear mixed model approaches (aside from perhaps the additional sex harmonization). Nonetheless, we now include the REGENIE 2.0.2 comparisons from our previous response as a supplemental table (Table S6).

The second reason is the scope of the paper. Although it presents some insight into the overlap of associations with drug targets, the main output of the paper is a database of PheWAS associations across a very wide range of phenotypes. Without a deeper examination of the role of these variants and the translational potential of these signals, this reads like a resource paper comparable to the Global Biobank Engine (McInnes et al., *Bioinformatics*, 2018, based on de Boever et al., *Nat Commun* 2018), or the unpublished Neale results. A paper of higher ambition should go further than, as the authors write, "expand the catalogue of statistically significant associations".

We feel the above summary is not a fair representation of the work described in the manuscript and appears to contrast with other comments received during this peer review process. Whilst de Boever et al (2018), McInnes et al (2018) (both pertaining to the Global Biobank Engine) and the unpublished Neale results might be considered useful resources, they do not describe findings from large-scale exome sequencing studies. We encourage the reviewer to draw from more recent comparison papers such as the Regeneron/GSK 50k exomes paper (PMID: 33087929) or the gnomAD papers (PMID: 32461654).

The paper would also benefit from more concise writing. In its current state it quite often gets lost in detail, reads rough and can seem unfocused at times. It would also be nice to make sure the results can be reproduced. Is the code available, have the authors packaged it in a way that allows reproduction and reuse, in particular with respect to statistical testing?

We have considerably edited the draft and worked through the Nature checklists to ensure our packages are public for reproduction/reuse.

I. 86. Although TOPMed is likely to improve imputation accuracy for RVs due to its size, HRC imputation doesn't allow to go too low in the allele frequency spectrum.

HRC is among the most common imputation panels adopted by the community and we believe it was rightfully cited. We have since removed this citation according to the reviewers suggestion.

I. 94-96. Please review the grammar of this sentence. This is the place to spend a bit more time introducing the various methods of aggregation and the statistical frameworks involved.

Due to space, we cannot expand such a methodological review. We leave the concise overview in the introduction and include details about other analytical frameworks in the corresponding methods sections. We cite our recent Nature Review Genetics paper (PMID: 31605095) throughout the manuscript as it provides a dedicated in-depth review of collapsing analysis frameworks and guidelines.

I. 193-198. The reason for this choice (computational efficiency) should be clearly stated from the start. The fact it turned out to be conservative doesn't need to be mentioned here.

We feel this is inaccurate for at least 2 reasons. Firstly, the choice of an exact test was driven by our considerable experience applying it to large-scale case-control sequencing studies, including our 2019 Nature Review Genetics paper (PMID: 31605095). In addition to the NRG paper, we have successfully published studies using this approach in Science (PMID: 25700176), Lancet Neurology (PMID: 28102150), NEJM (PMID: 33382938), JAMA Cardiology (PMID: 33326012) and multiple other journals. We provide some citations to these to further reaffirm that the selection wasn't based on any expected computational efficiency, but rather our experience. Secondly, we feel that not describing the full comparison would be withholding important information and we are not comfortable with such lack of transparency. Therefore, we retain our summary of the benchmarking results to SAIGE and REGENIE, which also clearly demonstrated that the Fisher's exact test was the more conservative test statistic – an important factor for many researchers.

I. 201-202. This observation indicates that the authors primarily conducted a frequency-agnostic GWAS, then noting that 26% of signals came from variants with MAF<0.5%. Focusing on rare variants from the get-go would be better. The authors can comment on the overlap between their common-variant signals and previously reported ones.

We decided to not withdraw any executed analyses from our paper. Our resubmission includes the frequency-agnostic analyses.

I. 214-216. Is this for rare significant PTVs/missense only or irrespective of MAF? How many of these gene/phenotype or indeed variant/phenotype relationships have been found by previous imputation-based UKB studies (e.g. Neale)? In general (not just for PTV/missense), a comparison with imputed GWAS in UKB is important both for novelty analysis and sensitivity. Perhaps a summary here with a couple of sentences and a more extended paragraph in the discussion?

Comparisons of imputation to sequencing have been provided in the Van Hout et al. Regeneron-GSK 50K Exomes paper (PMID: 33087929) and more recently in Barton et al. (2021) (PMID:34226706). We continue to believe that this is out of scope for our paper and we do not believe we will add anything beyond the previous literature focusing on this.

paragraph I. 227. This should be moved to the discussion. Remove runtime comparisons from the main text as the time advantage is expected and comes at the expense of robustness.

Our benchmarking comparisons demonstrate that our choice of statistic is not only more computationally efficient but also more robust. We thank the reviewer in particular for their previous suggestion to compare the genomic inflation factor distribution statistics across the various approaches which further endorses the selection of the exact test. We decide to retain the runtime comparisons (also requested by R2), and the lambda distributions across the three approaches in the

spirit of openness and transparency to the community. In that spirit, we have also included the REGENIE 2 comparisons in addition to SAIGE as part of the supplemental table 6.

I. 288-289. Remove "appropriate". Please reformulate "At this very..." by e.g. "Under this threshold, no positive associations are expected under the null."

We have revised the text accordingly (line 606-607).

I. 307. "in this study"? Do you support/recommend the use of MTR in general as a further filter to select variants more likely to be functional?

We have revised the text accordingly.

I. 370-372. I am still not convinced by this somatic variant paragraph. The added sentence does not particularly help. I suggest to substantially expand on these claims or take it out.

We have revised the text accordingly to simplify what we feel remains an interesting finding.

I. 435-442. Remove the suggestive association.

We have removed the suggestive association from the main text.

I. 441, 471. "insignificant" is not the proper wording. In Fig 4 legend "landed above" is also not serious wording.

We have revised the text accordingly.

Fig. 4. b. and d. seem quite surprising to me. I would expect a marked asymmetry between blue signals (not significant before inclusion of non-Europeans) and orange (used to be significant in Europeans only), with more of the former, and weaker delta-phred values for the latter. It seems very surprising that adding/removing ~5% of samples changes some p-values by a log-factor of 40?

As a reminder, this is a change of a log factor of 4 ($p=0.0001$) and not 40 as the Phred score is $10^-\log_{10}(p\text{-value})$. We have added an extra sentence in the figure legend to ensure this is not misinterpreted by readers.*

I 491-496. The proper way to test for this is through a conditional analysis.

We have consulted amongst the team and feel that this remains an appropriate analysis.

I 501-518. Emphasise that this was chosen to improve computational efficiency but is not something people should do in general.

This would not be an accurate reflection of our test statistic choice. Indeed the computational efficiency is a selection factor, but our decision is based on the many virtues of this approach – as described in our NRG 2019 paper (PMID: 31605095) and other respectable journals described earlier. We have, however, included the following clear points:

a) we clearly and openly outline the limitations and possible risks of adopting this method in similar studies and that all scientists should follow best experimental design and continue to be guided by important statistics such as genomic inflation factor in their decisions if appropriately applied – as they would with any other analytical framework.

b) we include an explicit recommendation that others use other tests (i.e. mixed linear models) in studies where they identify high genomic inflation in their test statistic distributions or where such careful quality control cannot be ensured.

c) we mention that more in-depth comparisons of the exact test and mixed linear models are required and should be performed in separate future studies, including studies introducing new frameworks should include the exact test as a comparator.

d) the use of an informed null distribution like the permutation-based and synonymous empirical null distribution adopted for collapsing analyses are both incredibly important approaches to defining study-wide p-value cut-offs in large multiple-phenotype settings like PheWAS. Researchers should continue to be diligent in their pre-association QC and cohort harmonisation, irrespective of statistical approach implemented, and continue to be guided by tried and tested genomic inflation factors in assessing the health checks of their large-scale genomics studies. Critically, the use of uninformed, arbitrary or less conservative p-value thresholds should be strongly discouraged.

I. 551. What method did the authors "introduce"?

We have revised the text accordingly.

I. 681. All software should be referenced. Rephrase I. 684-686 for clarity.

We have referenced all software and revised the text accordingly.

I. 701. Remove "to account for large case-control imbalances". Also remove "permit the study of extremely rare variants". If that were the reason the authors could have used this test only for those variants with $MAC < 7$.

We have removed these phrases accordingly.

I.710-712. Doesn't this contradict the "zero expected positives under the null"? In general the number of tests performed seems to change throughout the manuscript, but if the authors performed 108bn variant+4.5bn collapsing, the total exceeds the 38.7 tested, bringing the expected number (under permutation) to over 36 false positives?

This section refers to the ExWAS permutation run and we selected the dominant model whereas the empirical ExWAS studied 3 genetic models (108bn / 3). The collapsing was reported separately (see below) and in Table S20. We have revised the text accordingly.

I. 788. is this a second paragraph on n of 1 permutation?

This is the summary for the gene-based collapsing analyses.

I. 794. Combining it with I. 710, does this mean that 18 permuted p-values were between 1.9e-9 and 2e-9?

This confusion appears to arise from having performed the permutation analyses separately for the gene-based (collapsing) compared to the variant-level (exwas) analyses. We have clarified this text accordingly.